

**Distinct aerosol effects on cloud-to-ground lightning in the**
**plateau and basin regions of Sichuan, Southwest China**
Pengguo Zhao[1,2,3], Zhanqing Li[2], Hui Xiao[4], Fang Wu[5], Youtong Zheng[2],
Maureen C. Cribb[2], Xiaoai Jin[5], Yunjun Zhou[1]
[1]Plateau Atmosphere and Environment Key Laboratory of Sichuan Province, College
of Atmospheric Science, Chengdu University of Information Technology, Chengdu
610225, China
[2]Department of Atmospheric and Oceanic Science, Earth System Science
Interdisciplinary Center, University of Maryland, College Park, MD 20742, USA
[3]Key Laboratory for Cloud Physics of China Meteorological Administration, Beijing
100081, China
[4]Guangzhou Institute of Tropical and Marine Meteorology, China Meteorological
Administration, Guangzhou 510640, China
[5]State Laboratory of Remote Sensing Sciences, College of Global Change and Earth
System Science, Beijing Normal University, Beijing 100875, China
Correspondence: Zhanqing Li (zhanqing@umd.edu) and Pengguo Zhao
(zpg@cuit.edu.cn)





**Abstract.** The joint effects of aerosol, thermodynamic, and cloud-related factors on
cloud-to-ground lightning in Sichuan were investigated by a comprehensive analysis of
ground measurements made from 2005 to 2017 in combination with reanalysis data.
Data include aerosol optical depth, cloud-to-ground (CG) lightning density, convective
available potential energy (CAPE), mid-level relative humidity, lower- to mid-
tropospheric vertical wind shear, cloud-base height, total column liquid water (TCLW),
and total column ice water (TCIW). Results show that CG lightning density and
aerosols are positively correlated in the plateau region and negatively correlated in the
basin region. Sulfate aerosols are found to be more strongly associated with lightning
than total aerosols, so this study focuses on the role of sulfate aerosols in lightning
activity. In the plateau region, the lower aerosol concentration stimulates lightning
activity through microphysical effects. Increasing the aerosol loading reduces the cloud
droplet size, reducing the cloud droplet collision-coalescence efficiency and inhibiting
the warm-rain process. More small cloud droplets are transported above the freezing
level to participate in the freezing process, forming more ice particles and releasing
more latent heat during the freezing process. Thus, an increase in aerosol loading
increases CAPE, TCLW, and TCIW, stimulating CG lightning in the plateau region. In
the basin region, by contrast, the higher concentration of aerosols inhibits lightning
activity through the radiative effect. An increase in aerosol loading reduces the amount
of solar radiation reaching the ground, thereby lowering CAPE. The intensity of
convection decreases, resulting in less supercooled water transported to the freezing
level and fewer ice particles forming, thus increasing the total liquid water content.
Therefore, an increase in aerosol loading suppresses the intensity of convective activity
and CG lightning in the basin region.







## 1 Introduction


Aerosol-cloud-precipitation interactions are complicated, mainly reflected in the
influence of aerosols on cloud microphysical and radiation processes, i.e., aerosol-cloud
interactions (ACI) and aerosol-radiation interactions (ARI) (Rosenfeld et al., 2008;
Huang et al., 2009; Koren et al., 2014; Li et al., 2011, 2017, 2019; Oreopoulos et al.,
2020). The aerosol microphysical effect refers to the role of aerosols as cloud
condensation nuclei (CCN) and ice nuclei (IN), influencing the microphysical
processes of liquid- and ice-phase clouds. The aerosol radiation effect refers to the
absorption and scattering of solar radiation by aerosols, changing the radiation balance
between the atmosphere and the surface. The microphysical and radiative effects of
aerosols combined with dynamic processes influence weather and climate processes
through their links with meteorological conditions.
Lightning activity is mainly affected by atmospheric thermodynamic conditions
and is an important indicator of the development of convective systems. The collision
and separation of large and small ice particles mainly cause electrification. Supercooled
water, ice particles, and strong updrafts are the components needed for the occurrence
and development of lightning (MacGorman et al., 2001; Mansell et al., 2005; Williams,
2005; Price, 2013; Q. Wang et al., 2018; Qie and Zhang, 2019).
The differences in thermal conditions and aerosol loading between land and ocean
areas lead to a higher lightning frequency over land than over oceans (Williams and
Stanfill, 2002; Williams et al., 2004). Lightning activity over cities with higher aerosol
concentrations are more intense than that over clean suburbs (Westcott, 1995; Pinto et
al., 2004; Kar et al., 2009; Kar and Liou, 2014; Proestakis et al., 2016; Yair, 2018;
Tinmaker et al., 2019). An increase in aerosol concentration leads to the formation of
more small cloud droplets, which have difficulty forming raindrops due to their low
collision-coalescence efficiency, thus inhibiting the warm-rain process. These small
cloud droplets are transported above the freezing level, increasing the supercooled
water content in a thunderstorm and significantly enhancing the ice-phase process. The



freezing process releases more latent heat to stimulate convection, allowing more ice
particles to participate in the electrification process of collision and separation, thus
enhancing lightning activity (Khain et al., 2008; Mansell and Ziegler, 2013; P. Zhao et
al., 2015; Shi et al., 2015). A similar enhancement in lightning activity due to aerosols
was also found in oceanic regions, where aerosols and their precursors discharged by
ships significantly enhanced lightning activity over ship lanes (Thornton et al., 2017).
The influence of aerosols on thunderstorms is not linear. When the aerosol optical depth
(AOD) is less than 0.3, aerosols can stimulate lightning activity. However, the intensity
of lightning activity will be inhibited if the concentration of aerosols increases (Altaratz
et al., 2010; Stallins et al., 2013; X. Li et al., 2018; Q. Wang et al., 2018).
The effect of aerosols on convective clouds and lightning activity is not only
controlled by environmental factors, but also by aerosol type. Absorbing aerosols block
solar radiation from reaching the surface through radiative effects, which tends to
inhibit the development of convection. Hygroscopic aerosols can stimulate the
development of thunderstorms through microphysical effects under appropriate
environmental conditions (Wang et al., 2018). In central China, aerosol absorption of
solar radiation has increased the stability of the lower atmosphere, reducing
thunderstorm activity by 50% from 1961 to 2000 (Yang et al., 2013). In Nanjing in
eastern China, aerosols reduced the amount of solar radiation reaching the surface and
the convective available potential energy (CAPE), inhibiting the intensity of lightning
activity (Tan et al., 2016). In the Sichuan Basin, with its complex topography, the
influence of absorbing aerosols on strong convection is more complicated. During the
day, aerosols absorb solar radiation and increase the stability of the lower atmosphere,
accumulating a large amount of water vapor and energy in the basin. Under the
influence of the uplift of the mountain terrain at night, convection is excited, and
stronger convective precipitation is formed in the mountainous area (Fan et al., 2015).
In southeast China where the hygroscopicity of aerosols dominates, an increase in
aerosols in the plain areas significantly stimulates lightning activity (Yuan et al., 2011;
Y. Wang et al., 2011), while the influence of aerosols on thunderstorms in mountainous



areas with slightly higher altitudes is not prominent (Yang and Li, 2014). Aerosol
radiative and microphysical effects have different impacts on thunderstorms at different
stages of their development. In the Pearl River Delta region, the daytime radiative effect
delays lightning activity, while the aerosol microphysical effect at night further
stimulates lightning activity (Guo et al., 2016; Lee et al., 2016).
The eastern part of Sichuan province is a large basin, and the western part is the
easternmost part of the Tibetan Plateau. The thermal and moisture conditions in the
basin facilitate lightning activity (Xia et al., 2015; Yang et al., 2015). The Sichuan basin
is an area with high aerosol loading and with terrain not conducive to pollutant diffusion
(X. Zhang et al., 2012; L. Sun et al., 2016; Wei et al., 2019a, b). In this study, we
investigate the joint effects of aerosol, thermodynamic, and cloud-related conditions on
cloud-to-ground (CG) lightning activity under such special topographic conditions.
We mainly focus on the influence of aerosol, thermodynamic, and microphysical
factors on CG lightning density. Previous studies have suggested that aerosols affect
the intensity and polarity of lightning (Lyons et al., 1998; Naccarato et al., 2003; Carey
et al., 2007; Pawar et al., 2017). Future studies involving observational data analyses
and numerical simulations will investigate the mechanism by which aerosols affect the
lightning polarity by modulating the charge structure. This paper is organized as follows.
Section 2 describes the data and methodology used in the study. Section 3 presents and
discusses the results, and section 4 summarizes the study.

**2 Data and methodology**
**2.1 CG lightning**
Sichuan province is in southwest China, with the Qinghai-Tibet Plateau and
Hengduan Mountains to the west, the Qinba Mountains to the north, and the Yunnan-
Guizhou Plateau to the south (Fig. 1). The western part of Sichuan province is
dominated by plateau and mountainous terrain, with an average elevation of about 2000
to 4000 m, while the eastern part is dominated by a basin and hilly terrain, with an
average elevation of 300 to 700 m.





Hourly CG lightning flashes data from 2005 to 2017 were obtained from the
Sichuan Meteorological Bureau. CG lightning flashes are observed by the Sichuan
Lightning Detection Network (SLDN), which belongs to the China Lightning Detection
Network of the China Meteorological Administration (CMA), and consists of 25
detection sensors (Fig. 1). The average detection accuracy of the sensor is ~300 m, the
average detection radius is 300 km, and the detection efficiency is 80–90% (Yang et al.,
2015). The SLDN is based on the ground-based Advanced Time of Arrival and
Direction system, which uses improved accuracy from the combined technology
method (Cummins et al., 1998; CMA, 2009).
Positive CG lightning flashes with peak currents less than 15 kA are removed to
avoid the contamination of cloud-to-cloud lightning (Cummins and Murphy, 2006). A
flash is identified if the location of the first stroke is within 10 km, and the time interval
between two contiguous strokes is less than 0.5 seconds. If the polarity of the stroke is
different, it is a different flash (Cummins et al., 1998). To match the thermodynamic
and cloud-related parameters, the CG lightning data used in this study were calculated
at a 0.25° horizontal resolution. Many previous studies (e.g., Orville et al., 2011; Ramos
et al., 2011; Yang et al., 2015) have also discussed the basic characteristics of lightning
at a similar resolution.

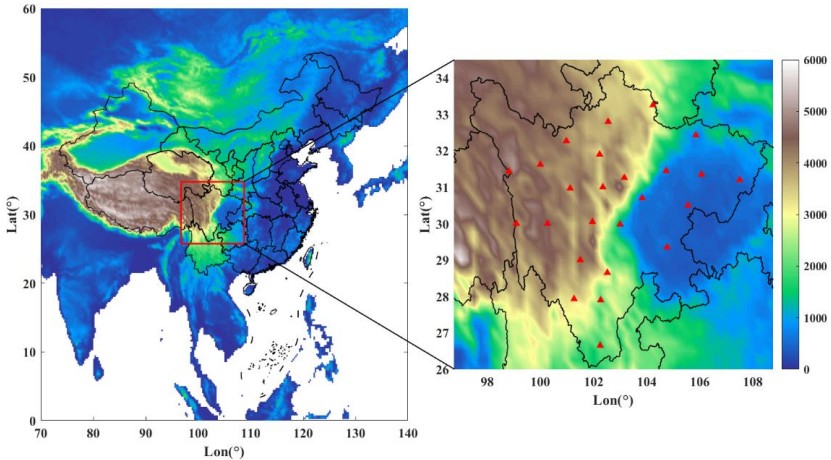


**Figure 1.** Location of Sichuan province with the color-shaded background showing



terrain heights (unit: m). The zoomed image shows the locations of the lightning sensors
(red triangles).

**2.2 AOD**

The Modern-Era Retrospective analysis for Research and Applications, version 2

(MERRA-2), dataset provided AODs from 2005 to 2017. The quality-controlled
MERRA-2 AOD product (at 550 nm) provides the optical thicknesses of different types
of aerosols, including total aerosol, sulfate, black carbon, organic carbon, and dust, with
a spatial resolution of $0.5° \times 0.625°$ (Randles et al., 2017; Buchard et al., 2017). To
match CG lightning data, we interpolated AOD data onto the same 0.25° spatial
resolution grid. The horizontal distribution and vertical structure of MERRA-2 aerosol
optical properties are in good agreement with satellite and aircraft observations
(Buchard et al., 2017). E. Sun et al. (2018, 2019) employed MODerate resolution
Imaging Spectroradiometer (MODIS) and Aerosol Robotic Network (AERONET)
AOD products to evaluate the MERRA-2 AOD over China. They reported that the
MERRA-2 and MODIS AODs agreed well and that the seasonal correlation coefficients
between the MERRA-2 and AERONET AODs ranged from 0.87 to 0.92.
**2.3 Thermodynamic and cloud-related parameters**

Thermodynamic and cloud-related factors include CAPE, mid-level relative

humidity (RH), lower- to mid-tropospheric vertical wind shear (SHEAR), cloud-base
height (CBH), total column liquid water (TCLW), and total column ice water (TCIW),
collected from ERA5 reanalysis data with a spatial resolution of $0.25° \times 0.25°$ (Dee et
al., 2011).

Hoffmann et al. (2019) indicated that the ERA5 reanalysis is more representative

of atmospheric convection, mesoscale cyclones, and mesoscale to synoptic-scale
atmospheric characteristics than the earlier ERA-Interim reanalysis. Freychet et al.
(2020) found that the dry-bulb temperature, wet-bulb temperature, and RH of the ERA5
reanalysis were representative through comparisons with ground observations made in
China. S. Lee et al. (2018) compared the water vapor and liquid water distributions
observed by a microwave radiometer in Seoul, South Korea, with that of the ERA5





reanalysis and found that they agreed well. Shou et al. (2019) confirmed that ERA5
data captured the cloud-top features based on multi-satellite observations made over the
Tibetan Plateau. Zhang et al. (2019) pointed out that the ERA5 precipitable water vapor
field agreed well with radiosonde and Global Navigation Satellite System observations.
Lei et al. (2020) examined the representation of ERA5 cloud-cover characteristics over
China through comparisons with satellite observations, reporting that (1) ERA5
overestimated the cloud cover by ~10%, and (2) the long-term trend in ERA5 cloud
cover was consistent with satellite observations. These studies suggest that ERA5
cloud-related data from China have sound quality.

CAPE is the most important factor controlling lightning, and climate projections

suggest that an increase in CAPE caused by global warming could increase global
lightning by 50% in the twenty-first century (Romps et al., 2014). The proxy composed
of precipitation rate and CAPE has a good correlation with observed lightning density
over the United States (Romps et al., 2018; Tippett and Koshak, 2018; Tippett et al.,
2019). CAPE is the factor with the highest relative contribution in various lightning
parameterization schemes (Bang and Zipser, 2016; Stolz et al., 2015, 2017).

Due to the large elevation fluctuation in Sichuan, pressure-level data are not

applicable to the analysis of the atmospheric vertical structure. So, pressure levels were
changed to geometric altitudes above ground level (AGL), using the barometric formula
(Minzner, 1977)

$$Z_2 = Z_1 + 18410 \left(1 + \frac{t_a}{273.15}\right) log \frac{P_1}{P_2}, \qquad (1)$$

where $Z_2$ and $Z_1$ are the elevations of the two isobaric levels (in m), $P_2$ and $P_1$ are the
pressures of the two isobaric levels (in hPa), $P_1$ is 1000 hPa, $Z_1$ is 0 m, and $t_a$ is the
average temperature of the two isobaric levels (in °C). The elevation minus topographic
height is the altitude AGL,

$$H = Z_2 - H_t, \qquad (2)$$

where $H$ is the geometric altitude AGL, and $H_t$ is the topographic height.

The mid-level RH and the lower- to mid-tropospheric SHEAR are important

humidity and dynamic parameters, directly affecting the formation, development,





propagation, and intensity of thunderstorms (Davies-Jones, 2002; Thompson et al.,
2007; Wall et al., 2014; Bang and Zipser, 2016). In this study, RH is the average RH in
the 3–5-km layer, and SHEAR is the vertical wind shear in the 0–5-km layer:
$$SHEAR = \sqrt{(u_2 - u_1)^2 + (v_2 - v_1)^2}, \qquad (3)$$
where $u_2$, $u_1$, $v_2$, and $v_1$ are zonal and meridional wind speeds at 5 km and 3 km,
respectively.
CBH, TCLW, and TCIW were selected to represent cloud-related parameters
affecting the development of lightning activity. CBH, negatively correlated with the
warm-cloud thickness, controls the convective structure and the polarity and intensity
of CG lightning by affecting the liquid water and ice water contents (Williams et al.,
2005; Carey and Buffalo, 2007; Stolz et al., 2017). Liquid water and ice water,
especially in the non-inductive electrification zone, directly control the processes of
charge generation and separation that determines the intensity of lightning of a
thunderstorm (Yair et al., 2010; Wong et al., 2013; Dafis et al., 2018).
In this study, we use Pearson correlation and partial correlation to discuss the
relationship between two elements at each grid point. Data from 156 months during the
period 2005–2017 were used, and monthly averages were calculated. Data at each grid
point were processed using a three-point moving average.
**3 Results and discussion**
**3.1 Distributions of CG lightning and AOD**
Due to the complex terrain in Sichuan, the CG lightning density and AOD differ
greatly across the province. The CG lightning density is highest over the basin region
in eastern Sichuan, with an annual average density of 1–3 flashes $km^{-2}$ $yr^{-1}$ (Fig. 2a).
The lightning density in western Sichuan is much lower than that in the basin region.
Yang et al. (2015) showed that the Sichuan basin is one of the most CG-lightning-active
regions in China, besides the Yangtze River Delta and the Pearl River Delta. The
dramatic difference in lightning density between the basin and the plateau stems
primarily from differences in humidity and thermal conditions. Another factor is the
generation of strong convective systems caused by the eastward migration of the



southwest vortex formed over the Tibetan Plateau to the basin area (Yu et al., 2007;
Zhang et al., 2014). The total AOD over the basin region is significantly higher than
that over the plateau region. The mean AOD over the basin is about 0.6–0.9, while that
over the plateau is about 0.15 (Fig. 2b). The aerosols in Sichuan are mainly composed
of sulfate aerosols, accounting for about 60–80% of the total AOD over the basin and
40–55% of the total AOD over the plateau (Fig. 2d). Aerosol concentrations over the
basin are higher than those over the plateau area, mainly because of the greater amount
of anthropogenic air pollutants emitted in the basin (Zhang et al., 2012). Also playing
important roles are the mountains around the basin and the low-pressure system at 700
hPa over the basin, resulting in a strong inversion above the planetary boundary layer
(Ning et al., 2018).

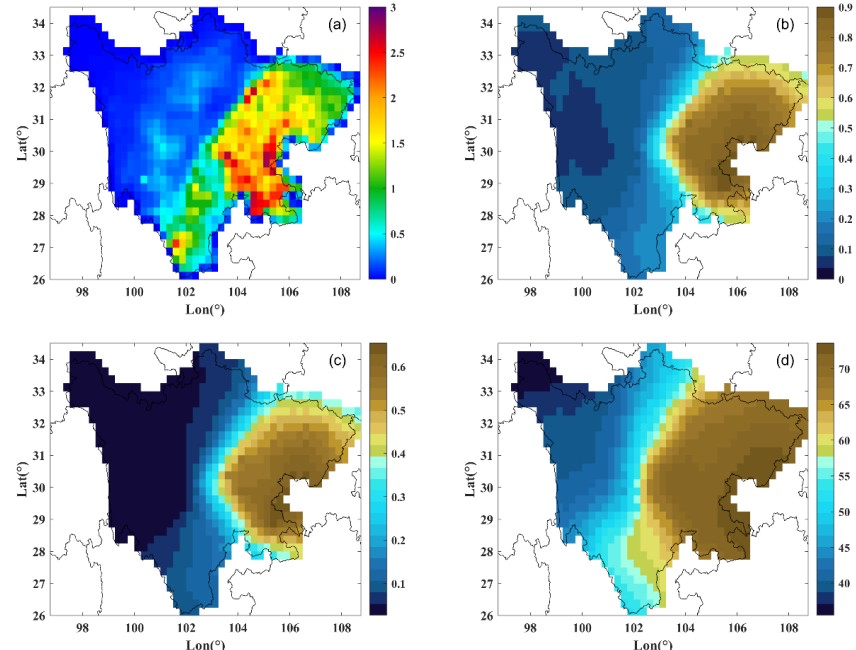

**Figure 2.** Distribution of (a) CG lightning density (unit: flashes km$^{-2}$ yr$^{-1}$), (b) total
AOD, (c) sulfate AOD, and (d) percentage of sulfate AOD in total AOD (unit: %) over
Sichuan.

**3.2 Correlation between AOD and CG lightning**
While the spatial patterns of lightning intensity (Fig. 2a) and AOD (Fig. 2b) bear





some resemblance, one cannot draw a straight conclusion that the latter is the cause of
the former because they are both influenced by the topography. However, the influences
of aerosols on lightning have been well established in previous studies by affecting the
local meteorological environment through aerosol radiative and microphysical effects
(Yang et al., 2013; Q. Wang et al., 2018; Z. Li et al., 2019). To circumvent the
topographic influence, Fig. 3 shows the Pearson correlation coefficients of total
AOD/sulfate AOD and CG lightning density in individual grid boxes in Sichuan. It is
interesting to note that the correlation between aerosol loading and lightning is opposite
in the plateau region and the basin region, i.e., a positive correlation in the plateau
region and a negative correlation in the basin region. This suggests that aerosols
stimulate lightning in the plateau region, but suppress lightning in the basin region.
Such a distinct difference may be related to differences in aerosol loading and local
environmental factors (Rosenfeld et al., 2007; Fan et al., 2009; Carrió and Cotton, 2014).
The maximum value of the positive correlation coefficient was about 0.5, occurring in
the plateau region of central Sichuan. The maximum values of the negative correlation
coefficients occurred in the basin region of eastern Sichuan. The absolute values of the
negative correlation coefficients are larger than those of the positive correlation
coefficients. The distribution of the correlation coefficients between lightning and
sulfate AOD is similar to that of total AOD, but there are more and larger positive
correlation coefficients than negative ones. Since sulfate AOD accounts for more than
80% of the total AOD in Sichuan, this study mainly discusses the relationship between
sulfate AOD and lightning activity.

Note that a statistical relationship between two variables does not necessarily

imply a true causality between the two for which much further insights are needed. The
spatial contrast exhibited in the correlation maps, however, conveys valuable
information about the causality because the influences of large-scale meteorology may
have little to do with the spatial pattern. The plateau and basin regions in this study are
outlined in Fig. 3 (right panel) to discuss the effects of sulfate aerosols on lightning
activity in the two regions separately.

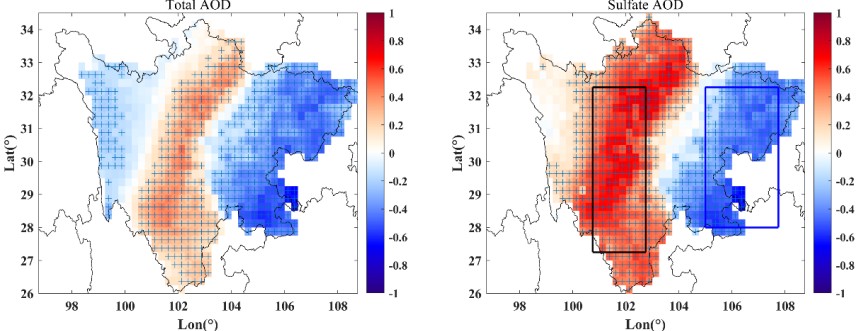

**Figure 3.** Pearson correlation coefficients between total AOD and CG lightning (left panel) and sulfate AOD and CG lightning (right panel) based on monthly data from 2005 to 2017. The correlation coefficient of each grid box is calculated from 156 monthly average datasets, and monthly average data are processed using a three-point moving average. Crosses in the figure indicate grid boxes that have passed the 95% significance test. The plateau region and the basin region are outlined by black and blue rectangles, respectively, in the right panel.

To further analyze the relationship between aerosols and lightning over Sichuan, Fig. 4 shows the CG lightning density as a function of sulfate AOD over the plateau and basin regions. Due to differences in emissions, the aerosol loading over the plateau region is much lighter than that over the basin region. The regional average sulfate AOD over the plateau region ranges from 0.03 to 0.15, and that over the basin region ranges from 0.22 to 0.87. The difference in CG lightning density is mainly related to the different meteorological conditions of the plateau and the basin. The monthly regional average CG lightning density over the plateau is $0.1 \times 10^{-3}$ to 0.35 flashes $km^{-2}$, while that over the basin is $0.1 \times 10^{-3}$ to 0.85 flashes $km^{-2}$. In the plateau region, the lightning density increases exponentially with increasing AOD, while in the basin region, the lightning density decreases exponentially with increasing AOD. This difference may be due to the different microphysical and radiative effects of different aerosol loadings. Previous studies (Koren et al., 2008, 2012; Altaratz et al., 2010, 2017) have noted a turning point of AOD = 0.3 with regard to the influence of AOD on clouds. For lower AOD, aerosols can stimulate lightning activity through microphysical effects. For higher AOD, aerosols reduce the solar radiation reaching the surface through the radiative effect, thus inhibiting lightning activity.



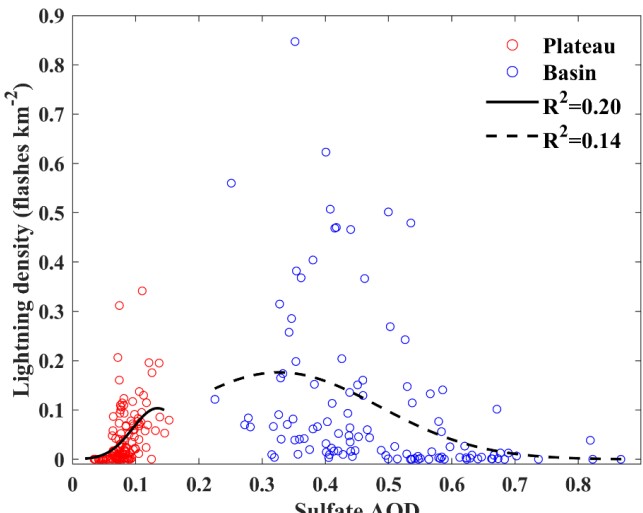

**Figure 4.** CG lightning density as a function of sulfate AOD over the basin (blue circles) and plateau (red circles) regions. Exponential-fit curves are shown, and coefficients of determination ($R^2$) are given.

### 3.3 Correlation between thermodynamic and cloud-related factors and CG lightning

Compared with the effect of aerosols on lightning activity, thermodynamic and cloud-related parameters are the decisive factors determining the occurrence and development of lightning activity (Williams, 2005; Williams et al., 2005; Saunders, 2008; Stolz et al., 2017). Figure 5 shows correlation coefficients between CAPE, RH, SHEAR, CBH, TCLW, and TCIW, and CG lightning density over Sichuan. The thermodynamic parameters CAPE and RH, especially CAPE, have significant excitation effects on lightning activity, while SHEAR shows a significant negative correlation with lightning. There is a positive correlation between TCIW and lightning density over Sichuan because the development of lightning mainly depends on the non-inductive electrification of the collision and separation of large and small ice particles. The more ice particles, the stronger the lightning activity will be. The correlation between CBH and lightning is opposite to that between TCLW and lightning in the plateau and basin regions. Over the plateau area, low cloud bases and high liquid water



contents are favorable for lightning activity, while over the basin, the opposite is seen.
A higher CBH means that the warm-cloud depth is thinner, so the liquid water content
will be less. In the plateau region, because of the compression effect of the plateau
topography on clouds, the warm-cloud depth is much thinner than that in the basin
region. Increasing a fixed amount of liquid water is conducive to transporting
supercooled water to the upper layer and promoting the development of the ice-phase
process. The more vigorous the ice-phase process is, the more intense the lightning
activity will be. Over the basin, where warm clouds are thicker, an increase in liquid
water will more likely promote the development of the warm-rain process rather than
the ice-phase process.

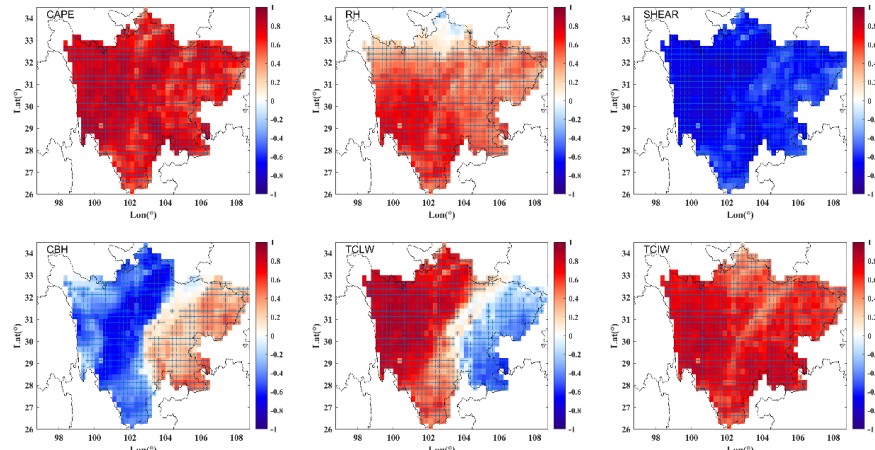

**Figure 5.** Pearson correlation coefficients between CAPE, RH, SHEAR, CBH, TCLW,
and TCIW and CG lightning. Crosses in the figure indicate grid boxes that have passed
the 95% significance test.

To avoid interactions between the factors involved and to discuss the relationships

between different factors and lightning activity more independently, Figs. 6 and 7,
respectively, show the partial correlation coefficients between thermodynamic and
cloud-related parameters and CG lightning density. In terms of the thermodynamic
parameters, the partial correlation coefficients show that the dependence of lightning
on RH and SHEAR is not significant. The partial correlation coefficient of some regions
in Sichuan is 0. Compared with RH, the absolute value of the negative partial





correlation coefficient of SHEAR is larger and more widely distributed, indicating that
SHEAR has a more significant impact (inhibition) on lightning activity than does RH.
CAPE is positively correlated with lightning in Sichuan, and the partial correlation
coefficient of many grid points is greater than 0.4, indicating that CAPE is a crucial
factor controlling lightning, as reported by others (Carey and Buffalo, 2007; Fuchs et
al., 2015; Bang and Zipser, 2016; Stolz et al., 2017).

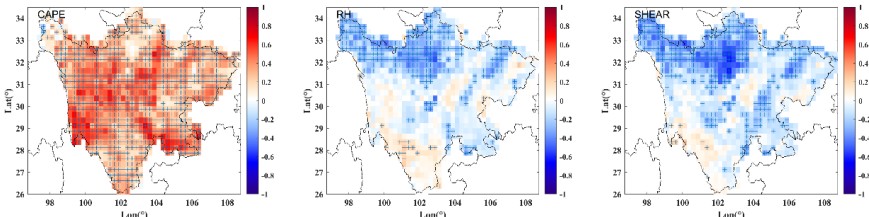

**Figure 6.** Partial correlation coefficients between CG lightning and thermodynamic
factors, i.e., CAPE, RH, and SHEAR. Crosses in the figure indicate grid boxes that have
passed the 95% significance test.

Among the cloud-related parameters, the partial correlation coefficients between

CBH and TCLW and lightning are lower, indicating that CBH and TCLW have less
significant influences on lightning density (Fig. 7). The existence of supercooled water
is one of the essential conditions for the electrification of thunderstorms. The
supercooled liquid water content in different temperature ranges can affect the polarity
of the charge carried by ice particles but cannot directly affect the intensity of the
electrical activity of thunderstorms (Saunders et al., 1991; Saunders, 2008). The
positive partial correlation coefficient between TCLW and lightning is relatively higher,
especially in the basin area, indicating that ice particles, as the carrier of charge, can
directly determine the occurrence and development process of lightning activity.

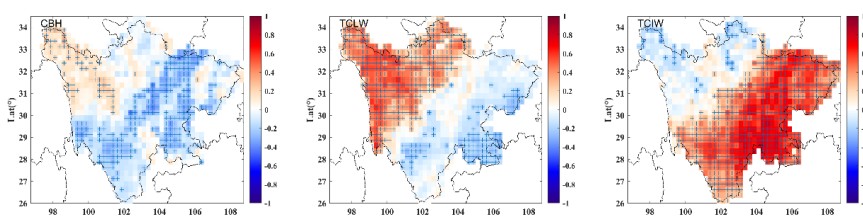

**Figure 7.** Partial correlation coefficients between CG lightning and cloud-related
factors, i.e., CBH, TCLW, and TCIW. Crosses in the figure indicate grid boxes that have



passed the 95% significance test.

To demonstrate the differences in thermodynamic and cloud-related factors
between the plateau and basin regions, Fig. 8 shows CG lightning density as a function
of the thermodynamic and cloud-related parameters in the plateau and basin regions,
based on monthly regionally averaged data. There is a significant positive correlation
between CAPE and CG lightning density in both the plateau and basin regions, with a
coefficient of determination ($R^2$) of 0.53 and 0.51, respectively. CAPE over the plateau
region is much smaller than that over the basin region. The maximum CAPE over the
plateau area is ~300 J kg$^{-1}$, while the maximum CAPE over the basin area is over 1000
J kg$^{-1}$. This is the main reason why the CG lightning density over the basin region is
larger than that over the plateau region. RH and CG lightning density were positively
correlated in both plateau and basin regions, but not significantly in the basin region
($R^2 = 0.08$). Due to the high altitude of the plateau and strong wind speeds there,
SHEAR in the plateau region (maximum value of 40 m s$^{-1}$) is significantly larger than
that in the basin region (maximum value of 15 m s$^{-1}$). The greater mid-level wind shear
over the plateau region suppresses the intensity of lightning activity.
Due to the compression of clouds by the plateau topography, the mean CBH over
the plateau region is relatively low, about 500–2000 m, while the mean CBH over the
basin region is about 1000–3500 m. The correlation between CBH and lightning density
is negative in the plateau. In the basin, however, there is barely any correlation ($R^2 =$
0.02). The much lower temperature over the plateau directly results in a lower liquid
water content there. The maximum value of TCLW is ~0.2 kg m$^{-2}$, while that in the
basin region is ~0.5 kg m$^{-2}$. Correlations in the plateau region are more significant than
in the basin region, with an $R^2$ of 0.26 and 0.19, respectively. The TCIWs over the
plateau and basin areas are similar in magnitude. The positive correlation between
TCIW and lightning density is also significant, with an $R^2$ of 0.34 and 0.42, respectively,
in the basin and plateau regions. Except for the correlation between CBH and lightning
in the basin region, the linear correlations between the other factors and lightning
passed the 95% significance test.



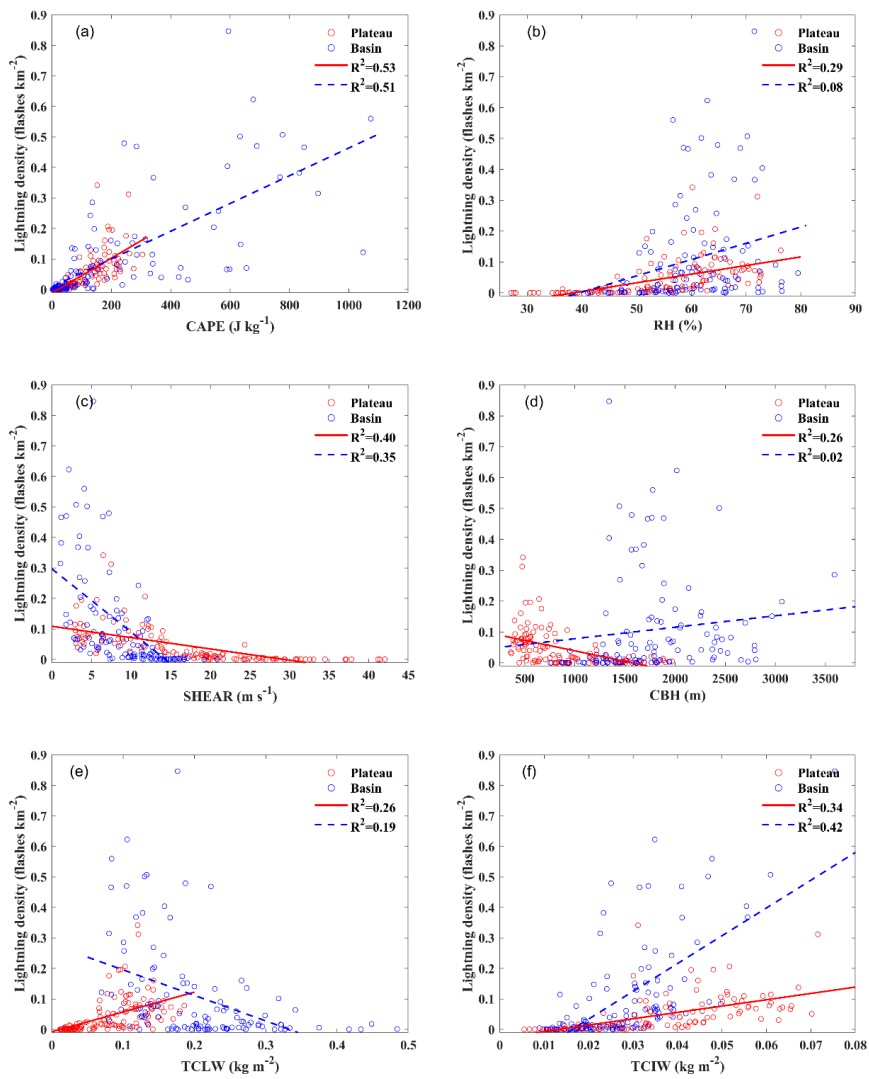

**Figure 8.** Lightning density as a function of thermodynamic and cloud-related factors in the basin (blue circles) and plateau (red circles) regions: (a) CAPE, (b) RH, (c) SHEAR, (d) CBH, (e) TCLW, and (f) TCIW. Linear-fit lines are shown, and coefficients of determination ($R^2$) are given.

**3.4 Joint effects of thermodynamic and cloud-related factors and aerosols on CG lightning**

Based on the partial correlation and linear fitting analyses, CAPE, SHEAR, TCLW, and TCIW are the main thermodynamic and cloud-related factors controlling CG





lightning over the Sichuan region. To analyze the joint effects of thermodynamic factors,
Figs. 9 and 10 show scatter plots between sulfate AOD, CAPE, SHEAR, TCLW, and
TCIW, and CG lightning in the plateau and basin regions. In the plateau region (Fig. 9),
increases in CAPE, TCLW, and TCIW enhance lightning activity. As discussed before
(Fig. 8), strong convective activity and more liquid water and ice water indicate that
strong updrafts transport a greater amount of liquid-phase and ice-phase particles to the
electrification area to participate in the electrification process, generating stronger
lightning activity. Aerosol excitation of lightning may be achieved by increasing CAPE,
TCLW, and TCIW. In the case of low aerosol loading, through ACI, an increase in
aerosols will reduce the size of cloud droplets and increase the concentration of cloud
droplets (Khain et al., 2008; Qian et al., 2009). Smaller cloud droplets reduce the
collision-coalescence efficiency and inhibit the warm-rain process. Small cloud
droplets that do not fall are transported above the freezing layer to participate in the
freezing process and release more latent heat. This is consistent with previous studies
(Mansell et al., 2013; P. Zhao et al., 2015; Altaratz et al., 2017; Fan et al., 2018; C. Zhao
et al., 2018) and explains the potential cause of the increase in aerosols, leading to an
increase in liquid water and ice water in thunderstorms, promoting convective activities.
From the joint influence of CAPE, SHEAR, TCLW, and TCIW on lightning activity
(bottom panels of Fig. 9), an increase in CAPE inhibits the vertical wind shear in the
lower to middle troposphere, which is conducive to the development of lightning
activity. Increasing CAPE also suggests that strong updrafts promote the development
of convection, resulting in the formation of more liquid water and ice water in the cloud.





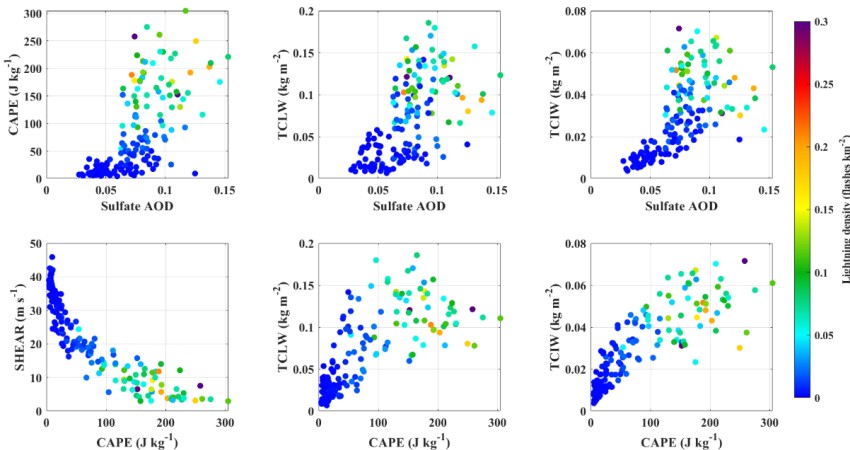


**Figure 9.** Joint effects of sulfate AOD, CAPE, SHEAR, TCLW, and TCIW on CG
lighting density over the plateau region. The color of the dots represents the CG
lightning density.


452   The aerosol loading over the basin region is much higher than that over the plateau

453 region, with sulfate AODs ranging from 0.2 to 0.9 (Fig. 10). Excessive aerosol loading

454 inhibits convective development through ARI. Aerosols reduce the solar radiation

455 reaching the surface through absorption and scattering, reducing the convective energy

456 of the surface and the lower atmosphere (Zhao et al., 2006; Jiang et al., 2018). Thus,

457 weak updrafts cannot transport liquid water above the freezing level. This may be why

458 the increase in aerosols leads to an increase in liquid water content and a decrease in

459 ice water content. Aerosols reduce the intensity of lightning activity by inhibiting the

460 development of convection and the formation of ice particles. This has also been

461 observed in other regions (Yang et al., 2013; Tan et al., 2016). CAPE is higher over the

462 basin region than over the plateau region (bottom panels of Fig. 11). An increase in

463 CAPE leads to a decrease in SHEAR and an increase in ice water content, promoting

464 the development of lightning, similar to the plateau region. Fan et al. (2009) found that

465 under large vertical wind shear conditions, an increase in aerosols inhibits the

466 development of convection. However, when CAPE exceeded 300 J kg$^{-1}$, an increase in

467 CAPE lead to a decrease in liquid water content. Convective clouds over the basin are

thicker than those over the plateau, and the high CAPE makes convection develop more
vigorously. In this way, liquid water is transported above the freezing level to participate
in the ice-phase process, forming more ice particles.

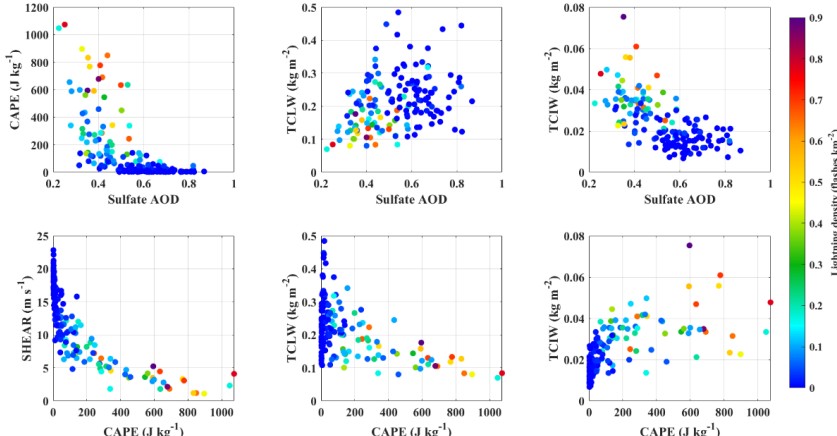


**Figure 10.** Same as in Fig. 9, but for the basin region.

According to the above, we hypothesize that the microphysical effect of aerosols

is responsible for stimulating lightning activity over the plateau region and that the
radiative effect of aerosols is responsible for suppressing lightning activity over the
basin region. The radiative effect of aerosols impacts lightning by affecting CAPE,
while the microphysical effect of aerosols impacts lightning by affecting the liquid
water and ice water contents. To further verify the radiative effect of aerosols in the
basin and the microphysical effect of aerosols in the plateau, two lightning sensitivity
parameters are employed:

$$RL_r = FC/CAPE,$$                                    (4)

where $RL_r$ is a relative measure of lightning sensitivity to the effect of CAPE, associated
with the aerosol radiative effect, and $FC$ is the CG lightning flash count. Tinmaker et
al. (2019) evaluated the impact of CAPE on lightning over land and oceanic regions by
using $FC/CAPE$:

$$RL_m = FC/(CAPE \times TCLW \times TCIW),$$          (5)

where $RL_m$ is a relative lightning parameter accounting for the effect of TCLW and





TCIW on lightning, associated with the aerosol microphysical effect. Since CAPE is an
essential factor for generating lightning, it is also considered in this formulation.

Figure 11 shows the Pearson correlation coefficients between $RL_r$, $RL_m$, and sulfate

AOD over Sichuan. Compared with the correlation between sulfate AOD and CG
lightning (right panel of Fig. 3), the negative correlation between sulfate AOD and $RL_r$
decreased significantly in the basin area, especially in the northern part of the basin,
while the positive correlation between AOD and $RL_r$ did not change significantly in the
plateau region. This suggests that the inhibitory effect of aerosols on lightning in the
basin region is dependent on the effect on CAPE, but not in the plateau region, which
also reflects the significant radiative effect of aerosols in the basin region. By
comparing the correlation between sulfate AOD and $RL_m$ (right panel of Fig. 11) and
the correlation between sulfate AOD and CG lightning (right panel of Fig. 3), the
positive correlation coefficients between sulfate AOD and $RL_m$ in the plateau region
decreased significantly, indicating that aerosols in the plateau region have a significant
microphysical effect, stimulating the development of lightning activity by influencing
liquid- and ice-phase particles in thunderstorms.

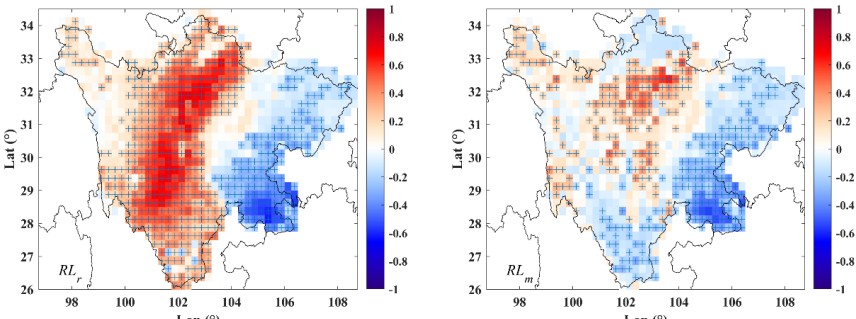

**Figure 11.** Same as in Fig. 3, but for $RL_r$ (left panel) and $RL_m$ (right panel).

**3.5 Multiple linear regression of CG lightning**

Because the physical processes involved in the development of lightning are

complex, many previous studies (e.g., Allen and Pickering, 2002; Tippett and Koshak,
2018) have parameterized lightning in weather and climate models by statistical



regression methods instead of describing the specific physical processes of lightning in
the model. Stolz et al. (2017) developed a global lightning parameterization scheme
based on multiple linear regression, combining aerosol and thermodynamic parameters,
which explained 69–81% of lightning activities in tropical and subtropical regions. The
multiple linear regression equations are based on the least-squares method and monthly
regionally averaged data. Since there is little or no lightning activity in winter, January,
February, and December are excluded. For the plateau region,
$Y = -0.023 + 0.52 \times 10^{-3} x_1 + 0.12 \times 10^{-3} x_2 - 6.01 \times 10^{-7} x_3 - 2.13 \times$
$10^{-5} x_4 - 0.62 x_5 - 0.14 x_6 + 0.06 x_7,$          (6)
and for the basin region,
$Y = -0.29 + 0.49 \times 10^{-3} x_1 - 0.25 \times 10^{-2} x_2 - 0.77 \times 10^{-2} x_3 - 1.53 \times 10^{-5} x_4 +$
$10.13 x_5 + 0.19 x_6 + 0.54 x_7,$          (7)
where $Y$ is the CG lightning density, $x_1$ is CAPE, $x_2$ is RH, $x_3$ is SHEAR, $x_4$ is CBH, $x_5$
is TCIW, $x_6$ is TCLW, and $x_7$ is sulfate AOD.

Figure 12 shows scatter plots and monthly distributions of CG lightning densities

from multiple linear regression and observations in the plateau and basin regions. The
scatter plots show that the modeled lightning density tends to be lower than the
observed lightning density. The correlation in the basin region ($R^2 = 0.66$) is higher than
that in the plateau region ($R^2 = 0.51$), but both are lower than the correlation reported
by Stolz et al. (2017). Note that Stolz et al. (2017) examined total lightning on a global
scale while this study focuses on CG lightning formed over a region with complex
terrain. The monthly distributions of observed and modeled CG lightning densities in
the plateau and basin regions show that multiple linear regression can reproduce the
seasonal variations in lightning activity well. Overall, the best agreement in both
regions is seen in summer. The best agreements in the plateau and basin regions are
seen in August and July, respectively.



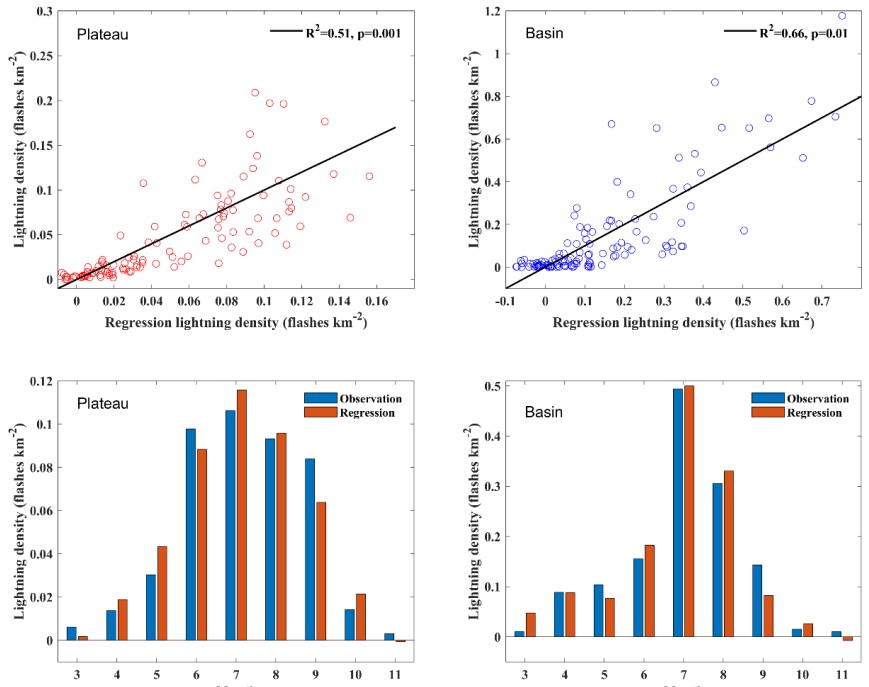

**Figure 12.** Scatter plots of observed CG lightning densities as a function of lightning densities from multiple linear regression in the plateau and basin regions (top panels) and their monthly distributions (bottom panels).

To further discuss the main impact factors that contribute to lightning, we use the stepwise regression method to select the top three impact factors. The stepwise regression equations based on the top three impact factors are as follows:

for the plateau region,

$$Y = -0.011 + 0.52 \times 10^{-3}x_1 + 0.25 \times 10^{-3}x_2 - 9.41 \times 10^{-6}x_4, \tag{8}$$

and for the basin region,

$$Y = -0.48 + 0.55 \times 10^{-3}x_1 + 9.35x_5 + 0.53x_7, \tag{9}$$

where $Y$ is the CG lightning density, $x_1$ is CAPE, $x_2$ is RH, $x_4$ is CBH, $x_5$ is TCIW, and $x_7$ is sulfate AOD. The top three factors contributing to lightning in the plateau region are CAPE, RH, and CBH, and the top three factors contributing to lightning in the basin region are CAPE, TCIW, and AOD, suggesting that aerosols have a more prominent effect on lightning in the basin region.

Figure 13 shows scatter plots and monthly distributions of CG lightning densities
from stepwise regression and observations in the plateau and basin regions. As seen in
Fig. 12, the modeled lightning density tends to be lower than the observed lightning
density, with $R^2$ values of 0.51 and 0.66 in the plateau and basin regions, respectively.
This also suggests that lightning activity can be reasonably modeled as long as factors
that contribute significantly to lightning, such as CAPE, are properly determined. The
monthly distributions of lightning densities modeled by stepwise regression agree with
observations from March to October reasonably well.

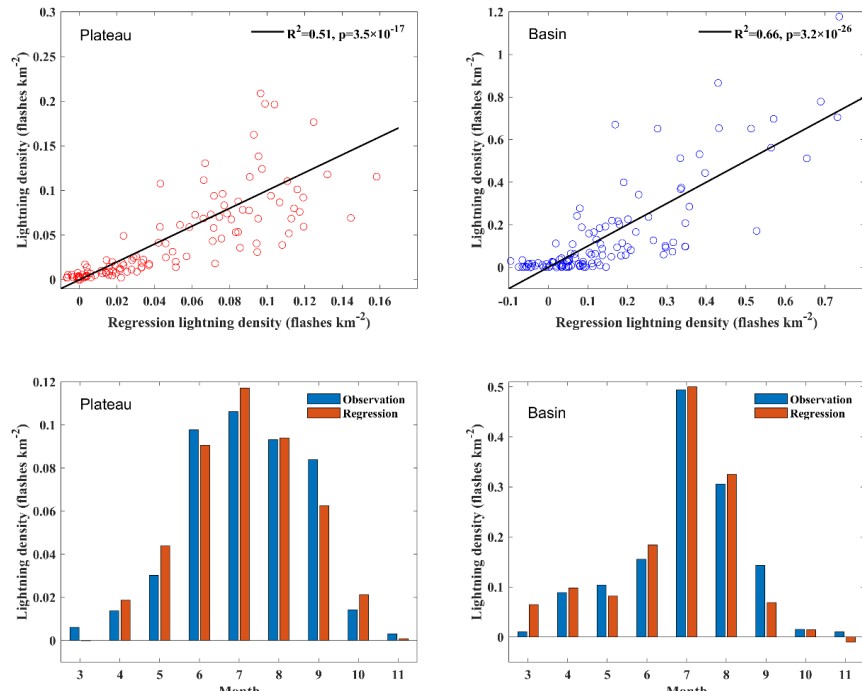

**Figure 13.** Scatter plots of observed CG lightning densities as a function of lightning
densities from stepwise regression in the plateau and basin regions (top panels) and
their monthly distributions (bottom panels).

**4 Conclusions**

In this study, we investigated the influence of aerosol, thermodynamic, and cloud-
related factors on CG lightning activity in the plateau and basin regions of Sichuan




province, a part of China with complex terrain. Data used to discuss the dependence of
the effect of aerosols on CG lightning on thermodynamic and cloud-related conditions
included the CG lightning density, sulfate AOD, CAPE, RH, SHEAR, CBH, TCLW,
and TCIW from 2005–2017.
CG lightning activity over the basin region was much more vigorous than that over
the plateau region, related to the thermodynamic difference between the two regions.
AODs in the basin region were also significantly higher than those in the plateau region,
mainly due to the large amounts of anthropogenic air pollutant emissions and the
mountainous terrain around the basin area that is not conducive to the diffusion of air
pollutants. CG lightning activity was positively correlated with AOD in the plateau
region, but negatively correlated with AOD in the basin region. The correlation between
sulfate AOD and lightning was stronger than that between total AOD and lightning, and
since sulfate AOD accounted for a high proportion of the total AOD, this study focused
on the role of sulfate AOD. The lightning density over the plateau region increased
exponentially with increasing AOD, while the lightning density over the basin region
decreased exponentially with increasing AOD.
CAPE, RH, and TCIW were significantly positively correlated with lightning
activity, while SHEAR was negatively correlated with lightning, suggesting that
convective uplift and ice-phase particles are essential factors for lightning activity. CBH
indirectly represents the warm-cloud thickness and is negatively correlated with TCLW.
The increase in TCLW in the plateau region is beneficial to lightning activity, but not
in the basin region, which may be related to the difference in warm-cloud depths
between the two regions. In the plateau region, because of the compression effect of the
plateau topography on clouds, warm clouds are very thin, and the high liquid water
content is conducive to conveying more supercooled water to the freezing level,
promoting the development of ice-phase clouds and lightning activity. In the basin
region, higher liquid water contents mean robust warm-cloud processes, which are more
conducive to the formation of warm rain than ice-phase processes, thus inhibiting
lightning activity. Partial correlation analyses indicate that CAPE, SHEAR, and TCIW





are important factors controlling lightning activity, especially CAPE.

To reveal the joint effects of aerosol, thermodynamic, and cloud-related factors on

CG lightning, AOD, CAPE, SHEAR, TCLW, TCIW were selected for further analysis.
In the plateau region, the aerosol loading is relatively low, stimulating lightning activity
through the microphysical effect. An increase in aerosol loading reduces the size of
cloud droplets, generating more but smaller cloud droplets, thus reducing the collision-
coalescence efficiency and inhibiting the warm-rain process. An increase in the liquid
water content of a cloud is conducive to the development of the ice-phase process,
which releases more latent heat and further stimulates convection. The increased
convection and the increase in ice particles lead to more intense lightning activity. In
the basin region, the aerosol loading is very high, which inhibits lightning activity
through the radiative effect. High concentrations of aerosols reduce the solar radiation
reaching the surface through absorption and scattering and reduce the convective
energy from the ground to the lower atmosphere. The weakening of the convective
uplift is not conducive to the transportation of liquid water above the freezing level and
inhibits the development of the ice-phase process. The weakening of convection and
the ice-phase process thus inhibits the intensity of lightning activity. The correlation
between $RL_m$ and AOD and the correlation between $RL_r$ and AOD further the idea that
aerosols over the plateau region affect the hydrometeor content in the atmosphere
through the microphysical effect, while aerosols over the basin region mainly affect
convective energy through the radiative effect, both of which affect lightning activity
differently.

*Data availability*. The CG lightning data can be obtained by contacting the first author
(zpg@cuit.edu.cn). MERRA-2 aerosol data can be download from
https://disc.sci.gsfc.nasa.gov/MERRA/ (last access: 9 September 2019), and the ERA5
data are from https://cds.climate.copernicus.eu/ (last access: 9 September 2019).

*Author contributions*. PZ and ZL designed the research ideas for this study; PZ carried


it out and prepared the manuscript; HX, Y. Zheng, FW, XJ, and Y. Zhou provided the
analysis ideas of meteorological and cloud-related parameters. MC edited the
manuscript.

*Competing interests*. The authors declare that they have no conflict of interest.

*Acknowledgments*. This research was jointly supported by the National Natural Science
Foundation of China (41905126, 41875169, 41705120), the National Key Research and
Development Project (2018YFC1505702), and the Key Laboratory for Cloud Physics
of China Meteorological Administration LCP/CMA (2017Z016). Pengguo Zhao
acknowledges China Scholarship Council for support (201808515075).

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
