# Peer review of "Distinct aerosol effects on cloud-to-ground lightning in the"

_Atmospheric Chemistry and Physics, 2020_

## Referee Comment (RC1) · Anonymous Referee #2 · 14 Jul 2020

This paper by Zhao et al. presented a detailed analysis of the aerosol effects on CG lightning in the plateau and basin regions of Sichuan, China, respectively. Results revealed that the aerosol effects on CG lightning are distinct between in the plateau and basin regions through microphysical effects and radiative effects, respectively. These findings can provide scientific insights for improving our understanding of the microphysical and radiative effects of aerosols on CG lightning in mountain-basin areas. Overall, this paper is well written and clearly describes the analysis, which addresses relevant scientific questions within the scope of ACP. I recommend this article to be published after these comments as below are addressed.

**Specific comments**

(1) From a general point of view, I would suggest the authors to maybe underline more efficiently the novelty of the study and its interest. Maybe that authors need, to do so, to modify the section of introduction. For instance, authors need to revise the aims of the study based on the results and conclusions. At least, it is necessary to highlight the different aerosol effects on CG lightning between in the plateau and basin regions of Sichuan, Southwest China.

(2) Both Lines 120-126 in the introduction section and Lines 138-143 in the Data and methodology section describe the complex topography around Sichuan province. Thus, I suggest that the authors move the contents of Lines 138-143 to the introduction section and rewrite the parts related to the complex topography around Sichuan Basin.

(3) Lines 128-132: "Previous studies have suggested that ···." belongs to future research plane and not to the research goal of this study, which is not suitable to appear in the section of introduction. These sentences should be moved to the discussion or conclusion section to indicate the limitations of this article that need to be solved in future research.

(4) Lines 275-279: the correlation between aerosol loading and lightning is negative in the basin region but is positive correlation in the plateau region. According to the above correlation coefficients, the authors concluded that aerosols stimulate lighting in the plateau region, but suppress lightning in the basin region. I think this conclusion is unconvincing. I thus suggest that the authors need to provide more sufficient evidence.

(5) Lines 288-289: 'Since sulfate AOD accounts for more than 80% of the total AOD in Sichuan, …', while as shown in Figure 2, sulfate AOD accounts for about 60-80% of the total AOD over the basin region and 40-55% of the total AOD over the plateau region. Please check it.

(6) I suggest that the authors need to perform a significance test on the curve fitting results in Figure 4.

(7) Lines 442-445: "From the joint ⋯, an increase in CAPE inhibits the vertical wind shear in the lower to middle troposphere⋯" Why does an increase in CAPE inhibits the vertical wind shear?

(8) In the sections 3.5, the multiple linear regressions of CG lighting have been developed in the plateau region (as shown in EQ.6) and in the basin region (as shown in EQ.7), respectively. However, the positive or negative values of the regression coefficients in front of each regression factor (such AOD, RH, CBH, TCIW, and TCLW) are inconsistent with the Pearson correlation coefficients between these factors and CG lightning in Figure 3 and Figure 5. For instance, the Pearson correlation coefficients between sulfate AOD and CG lightning are opposite between in the plateau region and the basin region; while the values of the regression coefficients associated with AOD are both positive in EQ.6 and EQ.7. I suggest authors to check the above results based on the multiple linear regression and give

reasonable explanations. In addition, the similar situations are also observed in EQ.9.

**Minor comments**

(1) It is better to give a table of acronym because there are many abbreviations in the manuscript.

(2) Line 123: 'diffusion' -> 'dispersion'

(3) Lines 122-124: "The Sichuan basin is an area with high areosol loading and with terrain … (X. Zhang et al., 2012; L. Sun et al., 2016; Wei et al., 2019a, b)" is suggested to be changed to "The Sichuan basin is an area with high areosol loading and with complex terrain … (Zhang et al., 2012; Sun et al., 2016; Wei et al., 2019a, b; Ning et al., 2017, 2019)".

(4) Line 127: 'influence'-> 'influences'

(5) Line 176: "E. Sun et al. (2018, 2019) employed …" -> "Sun et al. (2018, 2019) employed …"

(6) Line 192: "S. Lee et al. (2018) compared the …" -> "Lee et al. (2018) compared the …"

(7) Line 270: 'influenced'-> 'affected'

(8) Line 395: 'over 1000' -> 'greater 1000'

(9) Line 567: 'influence' -> 'influences'

(10) Line 577: 'diffusion' -> 'dispersion'

---

## Referee Comment (RC2) · Anonymous Referee #1 · 26 Jul 2020

The manuscript addresses a critical science question, i.e. how aerosol particles affect extreme weather phenomena like lightning. The ample amount of data from ground-based lightning network and reanalysis during 2005-2017 were analyzed to reveal the causal relationships between aerosol, lighting, and meteorological factors in Sichuan, China. This region in Southwest China is highly polluted but with large spatial variability of aerosols, and is understudied in previous researches. The findings about the aerosol microphysical invigoration effect over the plateau and the aerosol radiative suppression effect over the basin lend support to the notion about the non-linear (competing) aerosol effects on the deep convective systems. The study echoes the importance of taking aerosol effects into account for extreme weather analyses and predictions. I recommend its publication with ACP, while I also have comments below for the authors to address.

- L99-106, radiation absorption by aerosols can either suppress or enhance convection via altering CAPE depend on the heating vertical profile and the elevation where the convection initiates. Please see the discuss and the schematic of Wang et al. (2013, "New Directions: Light Absorbing Aerosols and Their Atmospheric Impacts").
- L178-179, it is not surprising to see good agreement between MERRA2-Aero and MODIS AOD, as MERRA2-Aero assimilates MODIS AOD product. Can the authors obtain the AOD from an independent satellite, such as MISR, to confirm the variability of the AOD near Sichuan?
- The uncertainty of cloud product from ERA5 over Southwest China seems unclear. Can the author make comparison of liquid/ice content between ERA5 and MODIS?
- Figs. 3,5-7. for the correlations between the time series of monthly mean data, do they mainly reflect the seasonality? Are they still significant if you remove the seasonality and look at anomalies (interannual variability) only?
- Figs, 6 and 7, how are the partial correlation coefficients calculated and how are they different from the total correlation coefficient? My understanding is the partial correlation is a measure of the dependence between two variables where the influence from other possible controlling variables (like meteorological parameters in this case) is removed. This method has been used in many previous aerosol-cloud studies (e.g. Zhao et al., 2019, *"Ice nucleation by aerosols from anthropogenic pollution"*). It seems the definition of partial correlation here is somewhat different with my understanding.
- L318-319, Liu et al. (2019, "Non-Monotonic Aerosol Effect on Precipitation in Convective Clouds over Tropical Oceans") examined satellite data and also reported a tipping point of precipitation response to aerosol perturbations, which occurs at AOD of 0.3.
- L330, please remove "Compared with the effect of aerosols on lightning activity", as there is no comparison in this sentence.

- Section 3.5 is confusing. The observed monthly and regional means of lightning density were used to build the multi-variate linear regression model. Then what's the point to compare the modeled lighting density with the observed one again? Please clarify.
- L574, please be specific what are the thermodynamic differences.

---

## Author Comment (AC1) · 5 Sep 2020

Dear editor and referee#1,

Thank you very much for your time and attentions on this work. The constructive comments and suggestions are very useful to improve our manuscript. Following are point-by-point responses to referee #1's comments. All the line numbers mentioned in responses are referred to the manuscript with changes marked.

(1) From a general point of view, I would suggest the authors to maybe underline more efficiently the novelty of the study and its interest. Maybe that authors need, to do so, to modify the section of introduction. For instance, authors need to revise the aims of the study based on the results and conclusions. At least, it is necessary to highlight the different aerosol effects on CG lightning between in the plateau and basin regions of Sichuan, Southwest China.

**Reply:** We have revised the introduction and highlighted the aim of the study. "There are significant geographical and environmental differences between the western Sichuan plateau and the eastern Sichuan basin. The thermal conditions of the western Sichuan Plateau are obviously weaker than those of the Sichuan Basin (Qie et al., 2003), and the aerosol concentration in the plateau is also significantly lower than that in the basin (Ning et al., 2018a). Previous studies (Yuan et al., 2011; Wang et al., 2011; Yang et al., 2013; Yang and Li, 2014; Fan et al., 2015) have suggested that aerosol effects on lightning activity differ significantly due to differences in topography and aerosol. The purpose of this study is to investigate any similarities and differences in the effects of aerosols on lightning activity in the context of different topography and aerosol concentrations between the Western Sichuan Plateau and Sichuan Basin." The details can be seen L136-145 of the revised manuscript.

(2) Both Lines 120-126 in the introduction section and Lines 138-143 in the Data and methodology section describe the complex topography around Sichuan province. Thus, I suggest that the authors move the contents of Lines 138-143 to the introduction section and rewrite the parts related to the complex topography around Sichuan Basin.

**Reply:** We have moved this sentence to the introduction section and rewrote the

parts related to the complex topography around Sichuan Basin. The details can be seen L125-130 of the revised manuscript.

(3) Lines 128-132: "Previous studies have suggested that …." belongs to future research plane and not to the research goal of this study, which is not suitable to appear in the section of introduction. These sentences should be moved to the discussion or conclusion section to indicate the limitations of this article that need to be solved in future research.

**Reply:** We have moved this sentence to the conclusion section to indicate the limitation of current study and the potential of the future study. The details can be seen L703-708 of the revised manuscript.

(4) Lines 275-279: the correlation between aerosol loading and lightning is negative in the basin region but is positive correlation in the plateau region. According to the above correlation coefficients, the authors concluded that aerosols stimulate lighting in the plateau region, but suppress lightning in the basin region. I think this conclusion is unconvincing. I thus suggest that the authors need to provide more sufficient evidence.

**Reply:** To further verify the stimulation and inhibition of aerosols on lightning activity and eliminate the interference of seasonality on the effects of aerosols on lightning, Pearson correlation coefficients between anomalies of total AOD and CG lightning and anomalies of sulfate AOD and CG lightning were implemented. As can be seen from the comparison between Fig. 3 and Fig. 4, the correlation coefficients between anomalies of AOD and lightning are significantly lower than those between AOD and lightning. While in an overall view, there is still a positive correlation between aerosols and lightning in the plateau region, and a negative correlation between aerosols and lightning in the basin region, especially for sulfate aerosols. This further verifies that aerosols have the potential to stimulate lightning activity in the plateau region and inhibit lightning activity in the basin region. The specific physical relationship will be further discussed below. The above discussion and the following figure as Figure 4 have been added to the revised manuscript. The details can be seen L344-364 of the revised

manuscript.

[Figure]

**Figure 4.** Pearson correlation coefficients between anomalies of total AOD and CG lightning (left panel) and anomalies of sulfate AOD and CG lightning (right panel) based on monthly data from 2005 to 2017. Crosses in the figure indicate grid boxes that have passed the 90% significance test.

(5) Lines 288-289: 'Since sulfate AOD accounts for more than 80% of the total AOD in Sichuan, …', while as shown in Figure 2, sulfate AOD accounts for about 60-80% of the total AOD over the basin region and 40-55% of the total AOD over the plateau region. Please check it.

**Reply:** It has been revised.

(6) I suggest that the authors need to perform a significance test on the curve fitting results in Figure 4.

**Reply:** We have carried out significance test on the curve fitting results by using F-test method, and the P value of both curves is less than 0.001, indicating that the curve fitting results are significant. We have redrawn Figure 4 as Figure 5, marked the P value in the figure, and made modifications in the revised manuscript. The details can be seen L384-387 of the revised manuscript.

[Figure]

**Figure 5.** CG lightning density as a function of sulfate AOD over the basin (blue circles) and plateau (red circles) regions. Exponential-fit curves are shown, and coefficients of determination ($R^2$) and p values are given.

(7) Lines 442-445:"From the joint …, an increase in CAPE inhibits the vertical wind shear in the lower to middle troposphere…" Why does an increase in CAPE inhibits the vertical wind shear?

**Reply:** CAPE is directly related to the upward movement, and CAPE can even be used to estimate the maximum updraft velocity (Molinari et al., 2012). Strong upward motion is not conducive to the development of vertical wind shear. previous studies (Li et al., 2013; Sherburn et al., 2016) on the complex of strong convection and mesoscale convection also found that the environmental vertical wind shear was smaller when CAPE was larger. Relevant references have been added to the revised manuscript. The details can be seen L524 of the revised manuscript.

(8) In the sections 3.5, the multiple linear regressions of CG lighting have been developed in the plateau region (as shown in EQ.6) and in the basin region (as shown in EQ.7), respectively. However, the positive or negative values of the regression coefficients in front of each regression factor (such AOD, RH, CBH, TCIW, and TCLW) are inconsistent with the Pearson correlation coefficients between these factors and CG

lightning in Figure 3 and Figure 5. For instance, the Pearson correlation coefficients between sulfate AOD and CG lightning are opposite between in the plateau region and the basin region; while the values of the regression coefficients associated with AOD are both positive in EQ.6 and EQ.7. I suggest authors to check the above results based on the multiple linear regression and give reasonable explanations. In addition, the similar situations are also observed in EQ.9.

**Reply:** In this study, we used multiple linear regression methods to fit the lightning density in Sichuan, and the regression factors included CAPE, RH, SHEAR, CBH, TCLW, TCIW, and AOD. In order to further analyze the most prominent factor contributing to the lightning density, we use the stepwise regression method to fit the lightning density. Because different factors contributed different proportions to the lightning density, there was a discrepancy between the positive and negative values of the regression factors and the positive and negative values of the Pearson correlation coefficient. Previous study (Wang et al., 2018) also had a similar situation.

**Minor comments**

(1) It is better to give a table of acronym because there are many abbreviations in the manuscript.

**Reply:** The acronym table has been added in the revised manuscript. The details can be seen in L274-276 in revised manuscript.

(2) Line 123:'diffusion' -> 'dispersion'

**Reply:** It has been revised.

(3) Lines 122-124: "The Sichuan basin is an area with high aerosol loading and with terrain … (X. Zhang et al., 2012; L. Sun et al., 2016; Wei et al., 2019a, b)" is suggested to be changed to "The Sichuan basin is an area with high aerosol loading and with complex terrain … (Zhang et al., 2012; Sun et al., 2016; Wei et al., 2019a, b; Ning et al., 2017, 2019)" .

**Reply:** It has been revised.

(4) Line 127: 'influence'-> 'influences'

**Reply:** It has been revised.

(5) Line 176:"E. Sun et al. (2018, 2019) employed …" ->"Sun et al. (2018, 2019) employed …"

**Reply:** It has been revised.

(6) Line 192:"S. Lee et al. (2018) compared the …" ->"Lee et al. (2018) compared the …"

**Reply:** It has been revised.

(7) Line 270: 'influenced'-> 'affected'

**Reply:** It has been revised.

(8) Line 395: 'over 1000' -> 'greater 1000'

**Reply:** It has been revised.

(9) Line 567: 'influence' -> 'influences'

**Reply:** It has been revised.

(10) Line 577:'diffusion' -> 'dispersion'

**Reply:** It has been revised.

Reference:

Fan, J., Rosenfeld, D., Yang, Y., Zhao, C., Leung, L. R. and Li, Z.: Substantial contribution of anthropogenic air pollution to catastrophic floods in Southwest China. Geophys. Res. Lett., 42(14), 6066-6075, https://doi.org/10.1002/2015GL064479, 2015.

Li, X., Guo, X. and Fu, D.: TRMM-retrieved cloud structure and evolution of MCSs over the northern South China Sea and impacts of CAPE and vertical wind shear, Adv. Atmos. Sci. 30, 77–88, https://doi.org/10.1007/s00376-012-2055-2, 2013.

Molinari, J., Romps, D.M., Vollaro, D. and Nguyen, L.: CAPE in tropical cyclones, J. Atmos. Sci., 69 (8): 2452–2463. https://doi.org/10.1175/JAS-D-11-0254.1, 2012.

Ning, G., Wang, S., Ma, M., Ni, C., Shang, Z., Wang, J. and Li, J.: Characteristics of air pollution in different zones of Sichuan Basin, China. Sci. Total Environ., 612, 975–984, https://doi.org/10.1016/j.scitotenv.2017.08.205, 2018a.

Qie, X., Toumi, R., Zhou, Y. J.: Lightning activity on the central Tibetan Plateau and its response to convective available potential energy, Chinese Science Bulletin,

48(3), 296–299, https://doi.org/10.1007/BF03183302, 2003.

Sherburn, K.D., Parker, M.D., King, J.R. and Lackmann, G.M.: Composite environments of severe and nonsevere high-shear, low-CAPE convective events, Wea. forecasting, 31(6), 1899-1927. https://doi.org/10.1175/WAF-D-16-0086.1, 2016.

Wang, Q., Li, Z., Guo, J., Zhao, C. and Cribb, M.: The climate impact of aerosols on the lightning flash rate: is it detectable from long-term measurements?, Atmos. Chem. Phys., 18(17), 12797–12816, https://doi.org/10.5194/acp-18-12797-2018, 2018.

Wang, Y., Wan, Q., Meng, W., Liao, F., Tan, H., and Zhang, R.: Long-term impacts of aerosols on precipitation and lightning over the Perl River Delta megacity area in China. Atmos. Chem. Phys., 11, 12421–12436, https://doi.org/10.5194/acp-11-12421-2011, 2011.

Yang, X., Yao, Z., Li, Z. and Fan, T.: Heavy air pollution suppresses summer thunderstorms in central China. J. Atmos. Sol.-Terr. Phy., 95, https://doi.org/10.1016/j.jastp.2012.12.023, 28–40, 2013.

Yang, X., and Li, Z.: Increases in thunderstorm activity and relationships with air pollution in southeast China, J. Geophys. Res. Atmos., 119, 1835–1844, https://doi.org/10.1002/2013JD021224, 2014.

Yuan, T., Remer, L. A., Pickering, K. E., Yun, H.: Observational evidence of aerosol enhancement of lightning activity and convective invigoration, Geophys. Res. Lett., 38, L04701, https://doi.org/10.1029/2010GL046052, 2011.

---

## Author Comment (AC2) · 5 Sep 2020

Dear editor and referee #2,

Thank you very much for your time and attentions on this work. The constructive comments and suggestions are very useful to improve our manuscript. Following are point-by-point responses to referee #2's comments. All the line numbers mentioned in responses are referred to the manuscript with changes marked.

(1) L99-106, radiation absorption by aerosols can either suppress or enhance convection via altering CAPE depend on the heating vertical profile and the elevation where the convection initiates. Please see the discuss and the schematic of Wang et al. (2013, "New Directions: Light Absorbing Aerosols and Their Atmospheric Impacts").

**Reply:** We have added this sentence "Absorbing aerosols in the boundary layer warm the atmosphere and cool the surface, which leads to the increase of atmospheric convective inhibition energy and the rise of convection condensation level (CCL), meanwhile the absorbing aerosols also leads to the increase of convective available potential energy above CCL. Once the lifting condition overcomes the convective inhibition energy, strong convective activity will be triggered (Wang et al., 2013)." and deleted the sentence "Absorbing aerosols block solar radiation from reaching the surface through radiative effects, which tends to inhibit the development of convection." The details can be seen L97-103 of the revised manuscript.

(2) L178-179, it is not surprising to see good agreement between MERRA2-Aero and MODIS AOD, as MERRA2-Aero assimilates MODIS AOD product. Can the authors obtain the AOD from an independent satellite, such as MISR, to confirm the variability of the AOD near Sichuan?

Reply: Figure S1 shows the spatial distribution of AOD based on the monthly data of MEERA2 and MISR data sets from 2005 to 2017. It can be seen from Fig S1 that the AOD spatial distribution of MISR is very close to that of MERRA, but the AOD value of MISR is smaller than that of MERRA2. Wei et al. (2019) suggest that there is a good consistency between MISR and MODIS AOD products in southwest China by using multi-satellite data comparison. The details can be seen L202-208 of the revised

manuscript.

[Figure]

**Figure S1.** The spatial distribution of annual mean AOD based on MERRA2 and MISR data sets from 2005 to 2017

(3) The uncertainty of cloud product from ERA5 over Southwest China seems unclear. Can the author make comparison of liquid/ice content between ERA5 and MODIS?

**Reply:** Due to the low spatial resolution (1°×1°) of MODIS monthly cloud product, we chose the cloud product of CLARA-A2 (0.25°×0.25°) for comparison with the cloud product of ERA5. CLARA-A2 is the second edition of the Satellite Application Facility on Climate Monitoring (CM SAF) cloud, albedo, and surface radiation dataset. The CLARA-A2 record provides cloud properties, surface albedo, and surface radiation parameters derived from the Advanced Very High-Resolution Radiometer (AVHRR) sensor (Karlsson et al., 2017; Karlsson and Håkansson, 2018). Figure S2 shows the spatial distribution of liquid water path (LWP) and ice water path (IWP) based on the monthly data of ERA5 and CLARA-A2 data sets from 2005 to 2015. LWP is high in the east and low in the west of Sichuan, while LWP in ERA5 is obviously lower than that of CLARA-A2 in the northwest of Sichuan. The spatial distribution of IWP in the two data sets are close, LWPs in northwestern Sichuan are higher than that in eastern and southern Sichuan.

We compared LWPs and IWPs of CLARA-A2 and ERA5 data sets, and overall, the cloud products of the two data sets were similar. For the continuity of data, LWP and IWP in ERA5 were selected in this study. We have added the above texts to the revised manuscript and the following figure to the supplement as figure S2. The details can be seen L230-236 of the revised manuscript.

[Figure]

**Figure S2.** The spatial distribution of annual mean LWP and IWP based on ERA5 and CLARA-A2 data sets from 2005 to 2015

(4) Figs. 3,5-7. for the correlations between the time series of monthly mean data, do they mainly reflect the seasonality? Are they still significant if you remove the seasonality and look at anomalies (interannual variability) only?

**Reply:** To eliminate the interference of seasonality on the effects of aerosols on lightning, Pearson correlation coefficients between anomalies of total AOD and CG lightning and anomalies of sulfate AOD and CG lightning were implemented. As can be seen from the comparison between Fig. 3 and Fig. 4, the correlation coefficients between the anomalies of AOD and lightning are significantly lower than those between AOD and lightning. While in an overall view, there is still a positive correlation between aerosols and lightning in the plateau region, and a negative correlation between aerosols and lightning in the basin region, especially for sulfate aerosols. This further verifies that aerosols have the potential to stimulate lightning activity in the plateau region and inhibit lightning activity in the basin region. The specific physical relationship will be further discussed below. The above discussion and the following figure as Figure 4 have

been added to the revised manuscript. The details can be seen L344-364 of the revised manuscript.

[Figure]

**Figure 4.** Pearson correlation coefficients between anomalies of total AOD and CG lightning (left panel) and anomalies of sulfate AOD and CG lightning (right panel) based on monthly data from 2005 to 2017. Crosses in the figure indicate grid boxes that have passed the 90% significance test.

Fig. S3 shows the correlation coefficients between the anomalies of CAPE, RH, SHEAR, CBH, TCLW, and TCIW and CG lightning. Compared with Figure 6 in the revised manuscript, the correlation coefficients are obviously smaller, especially in the basin region. The significances of the correlation between CG lightning and environmental factors are weakened, especially SHERA, CBH, and TCLW in the basin region. The above discussion has been added to the revised manuscript, and the following figure has been added to the supplement as Figure S3. The details can be seen L413-417 of the revised manuscript.

[Figure]

**Figure S3.** Pearson correlation coefficients between the anomalies of CAPE, RH, SHEAR, CBH, TCLW, and TCIW and CG lightning. Crosses in the figure indicate grid boxes that have passed the 95% significance test.

In the revised manuscript, we recalculated the partial correlation coefficient between meteorological factors and lightning, which is shown in Figure 7 in the revised manuscript. We used partial correlation coefficients to discuss the dependence of lightning on a meteorological factor relatively independently. The partial correlation coefficients in Figure 7 in the revised manuscript is small, while the partial correlation calculated by using the anomalies of variables is not significant.

(5) Figs, 6 and 7, how are the partial correlation coefficients calculated and how are they different from the total correlation coefficient? My understanding is the partial correlation is a measure of the dependence between two variables where the influence from other possible controlling variables (like meteorological parameters in this case) is removed. This method has been used in many previous aerosol-cloud studies (e.g. Zhao et al., 2019, "Ice nucleation by aerosols from anthropogenic pollution"). It seems the definition of partial correlation here is somewhat different with my understanding.

**Reply:** Figure 6 and Figure 7 in the original manuscript mainly aimed at analyzing the dependence of CG lightning on thermodynamic factors and cloud-related factors, so we analyzed the partial correlation between CG lightning and thermodynamic factors (CAPE, RH, and SHEAR) as well as lightning and cloud-related factors (CBH, TCLW, and TCIW), respectively. Based on your comment and Zhao et al. (2019), we recalculated the partial correlation coefficients between six meteorological factors (CAPE, RH, SHEAR, CBH, TCLW, and TCIW) and CG lightning in order to analyze the contribution of individual meteorological factor by eliminating the potential dependence on other meteorological factors. The corresponding discussion was modified, and the following figure was added to the revised manuscript as Figure 7. The details can be seen L436-449 of the revised manuscript.

[Figure]

**Figure 7.** Partial correlation coefficients between CG lightning and meteorological factors, i.e., CAPE, RH, SHEAR, CBH, TCLW. Crosses in the figure indicate grid boxes that have passed the 95% significance test.

(6) L318-319, Liu et al. (2019, "Non-Monotonic Aerosol Effect on Precipitation in Convective Clouds over Tropical Oceans") examined satellite data and also reported a tipping point of precipitation response to aerosol perturbations, which occurs at AOD of 0.3.

**Reply:** We have added this reference in the revised manuscript. The details can be seen L378 of the revised manuscript.

(7) L330, please remove "Compared with the effect of aerosols on lightning activity", as there is no comparison in this sentence.

**Reply:** It has been removed.

(8) Section 3.5 is confusing. The observed monthly and regional means of lightning density were used to build the multi-variate linear regression model. Then what's the point to compare the modeled lighting density with the observed one again? Please clarify.

**Reply:** In this study, we discussed the relationship between lightning density and

seven influence factors, including CAPE, RH, SHEAR, CBH, TCLW, TCIW, and AOD. We used Pearson correlation and partial correlation analysis methods to analyze the relative contributions of various influence factors to lightning activity. On this basis, we use multiple linear regression method and stepwise regression method to establish a model, which is used to test whether the seven influencing factors can reproduce the characteristics of lightning activity, and verify the influence factors that contribute more to lightning activity in the plateau and basin region. Previous study (Wang et al., 2018) also used similar methods to discuss the contribution of influence factors to lightning activity in Africa.

(9) L574, please be specific what are the thermodynamic differences.

**Reply:** The thermodynamic difference between the basin region and the plateau region mainly refers to the difference of CAPE. CAPE in the basin region is significantly higher than that in the plateau region, which leads to more vigorous lightning activity in the basin region (Qie et al., 2003). It has been revised accordingly in the revised manuscript. The details can be seen L655-656 of the revised manuscript.

**Reference:**

Karlsson, K. G., Anttila, K., Trentmann, J., Stengel, M., Meirink, J. F., Devasthale, A.: CLARA-A2: the second edition of the CM SAF cloud and radiation data record from 34 years of global AVHRR data. Atmos. Chem. Phys., 17, 5809–5828. https://doi.org/10.5194/acp-17-5809-2017, 2017.

Karlsson, K. G. and Håkansson, N.: Characterization of AVHRR global cloud detection sensitivity. Atmos. Chem. Phys., 11, 633–649. https://doi.org/10.5194/amt-11-633-2018, 2018.

Qie, X., Toumi, R., Zhou, Y. J.: Lightning activity on the central Tibetan Plateau and its response to convective available potential energy, Chinese Science Bulletin, 48(3), 296–299, https://doi.org/10.1007/BF03183302, 2003.

Wang, Y., Khalizov, A., Levy, M. and Zhang, R.: New Directions: Light absorbing aerosols and their atmospheric impacts. Atmos. Environ., 81, 713–715,

https://doi.org/10.1016/j.atmosenv.2013.09.034, 2013.

Wang, Q., Li, Z., Guo, J., Zhao, C. and Cribb, M.: The climate impact of aerosols on the lightning flash rate: is it detectable from long-term measurements?, Atmos. Chem. Phys., 18(17), 12797–12816, https://doi.org/10.5194/acp-18-12797-2018, 2018.

Wei, J., Peng, Y., Mahmood, R., Sun, L. and Guo, J., 2019. Intercomparison in spatial distributions and temporal trends derived from multi-source satellite aerosol products. Atmos. Chem. Phys., 19, 7183–7207, https://doi.org/10.5194/acp-19-7183-2019, 2019.

Zhao, B., Wang, Y., Gu, Y., Liou, K.N., Jiang, J.H., Fan, J., Liu, X., Lei, H., Yung, Y.L.: Ice nucleation by aerosols from anthropogenic pollution. Nat. Geosci., 12, 602–607, https://doi.org/10.1038/s41561-019-0389-4, 2019.

---

## Author Response (AR1)

Dear editor and referee#1,

Thank you very much for your time and attentions on this work. The constructive comments and suggestions are very useful to improve our manuscript. Following are point-by-point responses to referee #1's comments. All the line numbers mentioned in responses are referred to the manuscript with changes marked.

(1) From a general point of view, I would suggest the authors to maybe underline more efficiently the novelty of the study and its interest. Maybe that authors need, to do so, to modify the section of introduction. For instance, authors need to revise the aims of the study based on the results and conclusions. At least, it is necessary to highlight the different aerosol effects on CG lightning between in the plateau and basin regions of Sichuan, Southwest China.

**Reply:** We have revised the introduction and highlighted the aim of the study. "There are significant geographical and environmental differences between the western Sichuan plateau and the eastern Sichuan basin. The thermal conditions of the western Sichuan Plateau are obviously weaker than those of the Sichuan Basin (Qie et al., 2003), and the aerosol concentration in the plateau is also significantly lower than that in the basin (Ning et al., 2018a). Previous studies (Yuan et al., 2011; Wang et al., 2011; Yang et al., 2013; Yang and Li, 2014; Fan et al., 2015) have suggested that aerosol effects on lightning activity differ significantly due to differences in topography and aerosol. The purpose of this study is to investigate any similarities and differences in the effects of aerosols on lightning activity in the context of different topography and aerosol concentrations between the Western Sichuan Plateau and Sichuan Basin." The details can be seen L136-145 of the revised manuscript.

(2) Both Lines 120-126 in the introduction section and Lines 138-143 in the Data and methodology section describe the complex topography around Sichuan province. Thus, I suggest that the authors move the contents of Lines 138-143 to the introduction section and rewrite the parts related to the complex topography around Sichuan Basin.

**Reply:** We have moved this sentence to the introduction section and rewrote the parts related to the complex topography around Sichuan Basin. The details can be seen L125-130 of the revised manuscript.

(3) Lines 128-132: "Previous studies have suggested that …." belongs to future research plane and not to the research goal of this study, which is not suitable to appear in the section of introduction. These sentences should be moved to the discussion or conclusion section to indicate the limitations of this article that need to be solved in future research.

**Reply:** We have moved this sentence to the conclusion section to indicate the limitation of current study and the potential of the future study. The details can be seen L703-708 of the revised manuscript.

(4) Lines 275-279: the correlation between aerosol loading and lightning is negative in the basin region but is positive correlation in the plateau region. According to the above correlation coefficients, the authors concluded that aerosols stimulate lighting in the plateau region, but suppress lightning in the basin region. I think this conclusion is unconvincing. I thus suggest that the authors need to provide more sufficient evidence.

**Reply:** To further verify the stimulation and inhibition of aerosols on lightning activity and eliminate the interference of seasonality on the effects of aerosols on lightning, Pearson correlation coefficients between anomalies of total AOD and CG lightning and anomalies of sulfate AOD and CG lightning were implemented. As can be seen from the comparison between Fig. 3 and Fig. 4, the correlation coefficients between anomalies of AOD and lightning are significantly lower than those between AOD and lightning. While in an overall view, there is still a positive correlation between aerosols and lightning in the plateau region, and a negative correlation between aerosols and lightning in the basin region, especially for sulfate aerosols. This further verifies that aerosols have the potential to stimulate lightning activity in the plateau region and inhibit lightning activity in the basin region. The specific physical relationship will be further discussed below. The above discussion and the following figure as Figure 4 have been added to the revised manuscript. The details can be seen L344-364 of the revised manuscript.

[Figure]

**Figure 4.** Pearson correlation coefficients between anomalies of total AOD and CG lightning (left panel) and anomalies of sulfate AOD and CG lightning (right panel) based on monthly data from 2005 to 2017. Crosses in the figure indicate grid boxes that have passed the 90% significance test.

(5) Lines 288-289: 'Since sulfate AOD accounts for more than 80% of the total AOD in Sichuan, …', while as shown in Figure 2, sulfate AOD accounts for about 60-80% of the total AOD over the basin region and 40-55% of the total AOD over the plateau region. Please check it.

**Reply:** It has been revised.

(6) I suggest that the authors need to perform a significance test on the curve fitting results in Figure 4.

**Reply:** We have carried out significance test on the curve fitting results by using F-test method, and the P value of both curves is less than 0.001, indicating that the curve fitting results are significant. We have redrawn Figure 4 as Figure 5, marked the P value in the figure, and made modifications in the revised manuscript. The details can be seen L384-387 of the revised manuscript.

[Figure]

**Figure 5.** CG lightning density as a function of sulfate AOD over the basin (blue circles) and plateau (red circles) regions. Exponential-fit curves are shown, and coefficients of determination ($R^2$) and p values are given.

(7) Lines 442-445:"From the joint …, an increase in CAPE inhibits the vertical wind shear in the lower to middle troposphere…" Why does an increase in CAPE inhibits the vertical wind shear?

**Reply:** CAPE is directly related to the upward movement, and CAPE can even be used to estimate the maximum updraft velocity (Molinari et al., 2012). Strong upward motion is not conducive to the development of vertical wind shear. previous studies (Li et al., 2013; Sherburn et al., 2016) on the complex of strong convection and mesoscale convection also found that the environmental vertical wind shear was smaller when CAPE was larger. Relevant references have been added to the revised manuscript. The details can be seen L524 of the revised manuscript.

(8) In the sections 3.5, the multiple linear regressions of CG lighting have been developed in the plateau region (as shown in EQ.6) and in the basin region (as shown in EQ.7), respectively. However, the positive or negative values of the regression coefficients in front of each regression factor (such AOD, RH, CBH, TCIW, and TCLW) are inconsistent with the Pearson correlation coefficients between these factors and CG

lightning in Figure 3 and Figure 5. For instance, the Pearson correlation coefficients between sulfate AOD and CG lightning are opposite between in the plateau region and the basin region; while the values of the regression coefficients associated with AOD are both positive in EQ.6 and EQ.7. I suggest authors to check the above results based on the multiple linear regression and give reasonable explanations. In addition, the similar situations are also observed in EQ.9.

**Reply:** In this study, we used multiple linear regression methods to fit the lightning density in Sichuan, and the regression factors included CAPE, RH, SHEAR, CBH, TCLW, TCIW, and AOD. In order to further analyze the most prominent factor contributing to the lightning density, we use the stepwise regression method to fit the lightning density. Because different factors contributed different proportions to the lightning density, there was a discrepancy between the positive and negative values of the regression factors and the positive and negative values of the Pearson correlation coefficient. Previous study (Wang et al., 2018) also had a similar situation.

**Minor comments**

(1) It is better to give a table of acronym because there are many abbreviations in the manuscript.

**Reply:** The acronym table has been added in the revised manuscript. The details can be seen in L274-276 in revised manuscript.

(2) Line 123: 'diffusion' -> 'dispersion'

**Reply:** It has been revised.

(3) Lines 122-124: "The Sichuan basin is an area with high aerosol loading and with terrain … (X. Zhang et al., 2012; L. Sun et al., 2016; Wei et al., 2019a, b)" is suggested to be changed to "The Sichuan basin is an area with high aerosol loading and with complex terrain … (Zhang et al., 2012; Sun et al., 2016; Wei et al., 2019a, b; Ning et al., 2017, 2019)" .

**Reply:** It has been revised.

(4) Line 127: 'influence'-> 'influences'

**Reply:** It has been revised.

(5) Line 176:"E. Sun et al. (2018, 2019) employed …" ->"Sun et al. (2018, 2019) employed …"

**Reply:** It has been revised.

(6) Line 192:"S. Lee et al. (2018) compared the …" ->"Lee et al. (2018) compared the …"

**Reply:** It has been revised.

(7) Line 270: 'influenced'-> 'affected'

**Reply:** It has been revised.

(8) Line 395: 'over 1000' -> 'greater 1000'

**Reply:** It has been revised.

(9) Line 567: 'influence' -> 'influences'

**Reply:** It has been revised.

(10) Line 577:'diffusion' -> 'dispersion'

**Reply:** It has been revised.

[Figure]

**Figure S1.** The spatial distribution of annual mean AOD based on MERRA2 and MISR data sets from 2005 to 2017

(3) The uncertainty of cloud product from ERA5 over Southwest China seems unclear. Can the author make comparison of liquid/ice content between ERA5 and MODIS?

**Reply:** Due to the low spatial resolution (1°×1°) of MODIS monthly cloud product, we chose the cloud product of CLARA-A2 (0.25°×0.25°) for comparison with the cloud product of ERA5. CLARA-A2 is the second edition of the Satellite Application Facility on Climate Monitoring (CM SAF) cloud, albedo, and surface radiation dataset. The CLARA-A2 record provides cloud properties, surface albedo, and surface radiation parameters derived from the Advanced Very High-Resolution Radiometer (AVHRR) sensor (Karlsson et al., 2017; Karlsson and Håkansson, 2018). Figure S2 shows the spatial distribution of liquid water path (LWP) and ice water path (IWP) based on the monthly data of ERA5 and CLARA-A2 data sets from 2005 to 2015. LWP is high in the east and low in the west of Sichuan, while LWP in ERA5 is obviously lower than that of CLARA-A2 in the northwest of Sichuan. The spatial distribution of IWP in the two data sets are close, LWPs in northwestern Sichuan are higher than that in eastern and southern Sichuan.

We compared LWPs and IWPs of CLARA-A2 and ERA5 data sets, and overall, the cloud products of the two data sets were similar. For the continuity of data, LWP and IWP in ERA5 were selected in this study. We have added the above texts to the revised manuscript and the following figure to the supplement as figure S2. The details can be seen L230-236 of the revised manuscript.

[Figure]

**Figure S2.** The spatial distribution of annual mean LWP and IWP based on ERA5 and CLARA-A2 data sets from 2005 to 2015

(4) Figs. 3,5-7. for the correlations between the time series of monthly mean data, do they mainly reflect the seasonality? Are they still significant if you remove the seasonality and look at anomalies (interannual variability) only?

**Reply:** To eliminate the interference of seasonality on the effects of aerosols on lightning, Pearson correlation coefficients between anomalies of total AOD and CG lightning and anomalies of sulfate AOD and CG lightning were implemented. As can be seen from the comparison between Fig. 3 and Fig. 4, the correlation coefficients between the anomalies of AOD and lightning are significantly lower than those between AOD and lightning. While in an overall view, there is still a positive correlation between aerosols and lightning in the plateau region, and a negative correlation between aerosols and lightning in the basin region, especially for sulfate aerosols. This further verifies that aerosols have the potential to stimulate lightning activity in the plateau region and inhibit lightning activity in the basin region. The specific physical relationship will be further discussed below. The above discussion and the following figure as Figure 4 have been added to the revised manuscript. The details can be seen L344-364 of the revised manuscript.

[Figure]

**Figure 4.** Pearson correlation coefficients between anomalies of total AOD and CG lightning (left panel) and anomalies of sulfate AOD and CG lightning (right panel) based on monthly data from 2005 to 2017. Crosses in the figure indicate grid boxes that have passed the 90% significance test.

Fig. S3 shows the correlation coefficients between the anomalies of CAPE, RH, SHEAR, CBH, TCLW, and TCIW and CG lightning. Compared with Figure 6 in the revised manuscript, the correlation coefficients are obviously smaller, especially in the basin region. The significances of the correlation between CG lightning and environmental factors are weakened, especially SHERA, CBH, and TCLW in the basin region. The above discussion has been added to the revised manuscript, and the following figure has been added to the supplement as Figure S3. The details can be seen L413-417 of the revised manuscript.

[Figure]

**Figure S3.** Pearson correlation coefficients between the anomalies of CAPE, RH, SHEAR, CBH, TCLW, and TCIW and CG lightning. Crosses in the figure indicate grid boxes that have passed the 95% significance test.

In the revised manuscript, we recalculated the partial correlation coefficient between meteorological factors and lightning, which is shown in Figure 7 in the revised manuscript. We used partial correlation coefficients to discuss the dependence of lightning on a meteorological factor relatively independently. The partial correlation coefficients in Figure 7 in the revised manuscript is small, while the partial correlation calculated by using the anomalies of variables is not significant.

(5) Figs, 6 and 7, how are the partial correlation coefficients calculated and how are they different from the total correlation coefficient? My understanding is the partial correlation is a measure of the dependence between two variables where the influence from other possible controlling variables (like meteorological parameters in this case) is removed. This method has been used in many previous aerosol-cloud studies (e.g. Zhao et al., 2019, "Ice nucleation by aerosols from anthropogenic pollution"). It seems the definition of partial correlation here is somewhat different with my understanding.

**Reply:** Figure 6 and Figure 7 in the original manuscript mainly aimed at analyzing the dependence of CG lightning on thermodynamic factors and cloud-related factors, so we analyzed the partial correlation between CG lightning and thermodynamic factors (CAPE, RH, and SHEAR) as well as lightning and cloud-related factors (CBH, TCLW, and TCIW), respectively. Based on your comment and Zhao et al. (2019), we recalculated the partial correlation coefficients between six meteorological factors (CAPE, RH, SHEAR, CBH, TCLW, and TCIW) and CG lightning in order to analyze the contribution of individual meteorological factor by eliminating the potential dependence on other meteorological factors. The corresponding discussion was modified, and the following figure was added to the revised manuscript as Figure 7. The details can be seen L436-449 of the revised manuscript.

[Figure]

**Figure 7.** Partial correlation coefficients between CG lightning and meteorological factors, i.e., CAPE, RH, SHEAR, CBH, TCLW. Crosses in the figure indicate grid boxes that have passed the 95% significance test.

(6) L318-319, Liu et al. (2019, "Non-Monotonic Aerosol Effect on Precipitation in Convective Clouds over Tropical Oceans") examined satellite data and also reported a tipping point of precipitation response to aerosol perturbations, which occurs at AOD of 0.3.

**Reply:** We have added this reference in the revised manuscript. The details can be seen L378 of the revised manuscript.

(7) L330, please remove "Compared with the effect of aerosols on lightning activity", as there is no comparison in this sentence.

**Reply:** It has been removed.

(8) Section 3.5 is confusing. The observed monthly and regional means of lightning density were used to build the multi-variate linear regression model. Then what's the point to compare the modeled lighting density with the observed one again? Please clarify.

**Reply:** In this study, we discussed the relationship between lightning density and seven influence factors, including CAPE, RH, SHEAR, CBH, TCLW, TCIW, and AOD. We used Pearson correlation and partial correlation analysis methods to analyze the relative contributions of various influence factors to lightning activity. On this basis, we use multiple linear regression method and stepwise regression method to establish a model, which is used to test whether the seven influencing factors can reproduce the characteristics of lightning activity, and verify the influence factors that contribute more to lightning activity in the plateau and basin region. Previous study (Wang et al., 2018) also used similar methods to discuss the contribution of influence factors to lightning activity in Africa.

(9) L574, please be specific what are the thermodynamic differences.

  **Reply:** The thermodynamic difference between the basin region and the plateau region mainly refers to the difference of CAPE. CAPE in the basin region is significantly higher than that in the plateau region, which leads to more vigorous lightning activity in the basin region (Qie et al., 2003). It has been revised accordingly in the revised manuscript. The details can be seen L655-656 of the revised manuscript.

**Distinct aerosol effects on cloud-to-ground lightning in the**

**plateau and basin regions of Sichuan, Southwest China**

Pengguo Zhao[1,2,3], Zhanqing Li[2], Hui Xiao[4], Fang Wu[5], Youtong Zheng[2],

Maureen C. Cribb[2], Xiaoai Jin[5], Yunjun Zhou[1]

[revised manuscript text omitted]

There are significant geographical and environmental differences between the western Sichuan plateau and the eastern Sichuan basin. The thermal conditions of the western Sichuan Plateau are obviously weaker than those of the Sichuan Basin (Qie et al., 2003), and the aerosol concentration in the plateau is also significantly lower than that in the basin (Ning et al., 2018a). Previous studies (Yuan et al., 2011; Wang et al., 2011; Yang and Li, 2014; Fan et al., 2015) have suggested that aerosol effects on lightning activity differ significantly due to differences in topography and aerosol. The purpose of this study is to investigate any similarities and differences in the effects of aerosols on lightning activity in the context of different topography and aerosol
concentrations between the Western Sichuan Plateau and Sichuan Basin.

In this study, we investigate the joint effects of aerosol, thermodynamic, and cloud-
related conditions on cloud-to-ground (CG) lightning activity under such special
topographic conditions.

We mainly focus on the influence of aerosol, thermodynamic, and microphysical
factors on CG lightning density. Previous studies have suggested that aerosols affect
the intensity and polarity of lightning (Lyons et al., 1998; Naccarato et al., 2003; Carey
et al., 2007; Pawar et al., 2017). Future studies involving observational data analyses
and numerical simulations will investigate the mechanism by which aerosols affect the
lightning polarity by modulating the charge structure. This paper is organized as follows.
Section 2 describes the data and methodology used in the study. Section 3 presents and
discusses the results, and section 4 summarizes the study.

**2 Data and methodology**
**2.1 CG lightning**

[revised manuscript text omitted]

To further verify the stimulation and inhibition of aerosols on lightning activity and eliminate the interference of seasonality on the effects of aerosols on lightning, Pearson correlation coefficients between anomalies of total AOD and CG lightning and anomalies of sulfate AOD and CG lightning were implemented. As can be seen from the comparison between Fig. 3 and Fig. 4, the correlation coefficients between the anomalies of AOD and lightning are significantly lower than those between AOD and lightning. While in an overall view, there is still a positive correlation between aerosols and lightning in the plateau region, and a negative correlation between aerosols and lightning in the basin region, especially for sulfate aerosols. This further verifies that aerosols have the potential to stimulate lightning activity in the plateau region and inhibit lightning activity in the basin region. The specific physical relationship will be further discussed below.

Note that a statistical relationship between two variables does not necessarily imply a true causality between the two for which much further insights are needed. The spatial contrast exhibited in the correlation maps, however, conveys valuable information about the causality because the influences of large-scale meteorology may have little to do with the spatial pattern.

[Figure]

**Figure 4.** Pearson correlation coefficients between anomalies of total AOD and CG lightning (left panel) and anomalies of sulfate AOD and CG lightning (right panel) based on monthly data from 2005 to 2017. Crosses in the figure indicate grid boxes that have passed the 90% significance test.

[revised manuscript text omitted]

Ning, G., Wang, S., Ma, M., Ni, C., Shang, Z., Wang, J. and Li, J.: Characteristics of air pollution in different zones of Sichuan Basin, China. Sci. Total Environ., 612,

975–984, https://doi.org/10.1016/j.scitotenv.2017.08.205, 2018a.

Ning, G., Wang, S., Yim, S., Li, J., Hu, Y., Shang, Z., Wang, J. and Wang, J.: Impact of low-pressure systems on winter heavy air pollution in the northwest Sichuan Basin, China, Atmos. Chem. Phys., 18(18), 13601–13615, https://doi.org/10.5194/acp-18-13601-2018, 2018b.

Ning, G., Yim, S.H.L., Wang, S., Duan, B., Nie, C., Yang, X., Wang, J. and Shang, K.: Synergistic effects of synoptic weather patterns and topography on air quality: a case of the Sichuan Basin of China, Clim. Dynam., 53(11), 6729–6744, https://doi.org/10.1007/s00382-019-04954-3, 2019.

[revised manuscript text omitted]